# Learning 3D Hypersonic Flow with Physics-Enhanced Neural Fields: A Case Study on the Orion Reentry Capsule

**Haitz Sáez de Ocáriz Borde** *Ratio Labs*

**Pietro Innocenzi** *Ratio Labs, Imperial College London*

**Flavio Savarino** *Ratio Labs*

**Andrei Cristian Popescu** *Ratio Labs*

**Pantelis Papageorgiou** *Ratio Labs*

**Reviewed on OpenReview:** *https://openreview.net/forum?id=ce2X1X3l0Y*

## Abstract

We develop a 3D aerothermodynamic surrogate for the Orion reentry capsule at hypersonic speeds, a timely case study given its role in upcoming lunar missions. The large computational meshes required for these scenarios make traditional computational fluid dynamics impractical for full-mission performance prediction and control. In this work, we propose physics-enhanced 3D neural fields for predicting steady hypersonic flow around aerodynamic bodies. The model maps spatial coordinates and angle of attack to pressure, temperature, and velocity components. We enhance the base model with Fourier positional feature mappings, which allow it to capture the sharp discontinuities typical of hypersonic flows, and further constrain the solution by imposing no-slip and isothermal wall conditions. We compare our proposed approach to other surrogate alternatives, such as graph neural networks, and demonstrate its superior performance in capturing the steep gradients ubiquitous in this regime. Our formulation yields a continuous and computationally efficient aerothermodynamic surrogate that supports rapid exploration of operating conditions based on angle of attack variation under realistic flight profiles (for a fixed capsule geometry). While we focus on Orion, the proposed framework provides a general methodology for data-driven simulation in 3D hypersonic aerothermodynamics.

## 1 Introduction

Reynolds-Averaged Navier-Stokes (RANS), Large Eddy Simulation (LES), and Direct Numerical Simulations (DNS) are all Computational Fluid Dynamics (CFD) approaches (Roache, 1972; Anderson, 1995; Tu et al., 2018) that approximate the Navier-Stokes equations (Navier, 1822; Stokes, 1845) governing fluid flows. Fidelity depends on purpose, with DNS for detailed research physics and RANS for most industrial engineering, since DNS becomes extremely expensive beyond simple geometries and engineers must trade accuracy for computational feasibility (Pope, 2000). Artificial neural networks offer fast simulation and rapid prototyping. Trained on CFD data, they can produce solutions orders of magnitude faster than traditional CFD. Because numerical simulations already have error, neural networks add another layer of uncertainty as approximations. This trade-off can be acceptable for early exploration of geometry and flow design configurations in aircraft and spacecraft development, especially considering that our neural network can predict fields that take hundreds of hours to converge using traditional CFD (Section 4.6) on the order of seconds (Section 7.5). Once promising candidates are identified, higher-resolution CFD simulations can refine the results, followed by wind-tunnel testing. In this way, the design process could follow a natural progression: rapid neural exploration, targeted CFD simulation, and eventual experimental testing.

In this work, we develop a 3D aerothermodynamic surrogate using the Orion reentry capsule as a test case. We focus on this configuration due to its high degree of symmetry, which makes it well suited for an initial study, as well as its relevance to upcoming lunar missions and real-world applications. Our contributions are as follows:

- We generate an ML-ready 3D CFD dataset for the Orion reentry capsule, sweeping the angle of attack (AoA) and utilizing advanced mesh refinement techniques to accurately model the hypersonic bow shock.

- We propose the use of physics-enhanced 3D neural fields to predict steady hypersonic flow around the geometry (note that we focus on boundary conditions, but do not include conservation laws or PDE residuals).

- We perform a thorough empirical study comparing neural fields and graph neural networks (GNNs) in the context of hypersonic flow modeling. This helps shed light on strengths and weaknesses of different learning methods.

- We conduct an in-depth aerothermodynamic analysis after training our models, evaluating performance not only using machine learning metrics but also from an aerodynamicist's perspective.

In summary, this work is best understood as an application-focused empirical study and dataset contribution for 3D hypersonic aerothermodynamic surrogate modeling, presented through a case study on the Orion reentry capsule.

## 2  Related work

In aerodynamics, reduced-order models (ROMs) are simplified mathematical representations designed to capture the essential behavior of complex aerodynamic systems while significantly reducing computational cost (Ripepi et al., 2018). Within the literature, both unsteady laminar and turbulent flow regimes have been studied using methods that extract coherent structures and flow patterns to inform dynamical models with predictive and control capabilities (Durmaz et al., 2013; Stabile et al., 2017; Guzman-Inigo et al., 2019; Nidhan et al., 2020; Schmidt et al., 2018; Fukami et al., 2021; Giannopoulos & Aider, 2020). Linear models have shown good performance even in nonlinear and highly nonlinear, large-scale systems; however, they typically require real-time, partial measurements of the full system state to account for nonlinear effects (Guzman-Inigo et al., 2019; Mikhaylov et al., 2021; Loiseau et al., 2018; Savarino & Papadakis, 2022). In contrast, nonlinear models rely on more sophisticated architectures but often suffer from tuning difficulties and reduced interpretability (Nair & Goza, 2020; Kim et al., 2021; Rozov & Breitsamter, 2021).

In parallel, deep learning approaches have been increasingly applied to a wide range of aerospace engineering problems, including turbulence modeling, aerodynamic shape optimization, wall-flux-based wall models, and rocket liquid engine design, to name a few (Ling et al., 2016; Borde et al., 2021a;b; Sayyari et al., 2022; Fang et al., 2019; Kaandorp, 2018; Li et al., 2022; Xu et al., 2021; Li & Zhang, 2021; Liu et al., 2023; Lozano-Duran & Bae, 2020; Waxenegger-Wilfing et al., 2021). Deep learning methods are capable of approximating complex input-output relationships and capturing highly nonlinear dynamics directly from data. Previous aerothermodynamic surrogates include those by Borde et al. (2023) for 2D flows around rocket geometries.

## 3  Preliminaries for Compressible Laminar Flow Simulations

To make the subsequent data-generation procedure self-contained, we first review the governing flow equations and modeling assumptions used in the CFD simulations. CFD encompasses a hierarchy of mathematical models whose appropriate selection depends on the physical regime, geometric complexity, and accuracy required of the target application. Two modeling axes structure this hierarchy: the treatment of compressibility, and the resolution of turbulent scales. Flows are typically treated as effectively incompressible when the Mach number satisfies $M \lesssim 0.3$, in which case density variations may be neglected and the governing

equations reduce to a divergence-free velocity field with decoupled thermodynamics; otherwise, the fully compressible Navier-Stokes equations must be retained to capture compressible phenomena such as shock waves, expansion fans, and aerodynamic heating (Anderson, 2003). With respect to turbulence, available frameworks span from RANS, which models the effects of the entire turbulent spectrum via closure models, through hybrid Detached Eddy Simulation (DES) and LES, which resolve the energy-containing eddies and model only the unresolved subgrid scales, to DNS, which resolves every turbulent length and time scale without modeling assumptions (Pope, 2000; Slotnick et al., 2014). Increasing fidelity is accompanied by a steeply rising computational cost: the grid-point requirement for wall-resolved DNS of a turbulent boundary layer scales approximately as $Re_L^{2.05}$ to $Re_L^{2.64}$, rendering its application to fully three-dimensional industrial geometries intractable on present-day hardware (Choi & Moin, 2012; Slotnick et al., 2014).

At an altitude of approximately 50 kft, the flow over a hypersonic capsule may exhibit laminar, transitional, or turbulent behavior depending on the local Reynolds number, surface conditions, and vehicle geometry. In the present work, the flowfield is assumed to be fully laminar in order to reduce the complexity of the initial dataset used to train the neural field representation. This assumption is adopted primarily as a simplification for the proof-of-concept study and does not constitute a limitation of the proposed methodology. The neural field framework is largely agnostic to the underlying flow model and can readily accommodate additional variables associated with turbulence closure models with only a modest increase in computational cost. Consequently, there is no inherent restriction preventing the application of the proposed methodology to RANS simulations. Extension to turbulent flowfields is therefore considered a straightforward direction for future work. The governing equations for this regime are the compressible Navier-Stokes equations in their laminar form. For a viscous, heat-conducting ideal gas these read, in conservative form:

$$\frac{\partial \rho}{\partial t} + \frac{\partial \rho u_i}{\partial x_i} = 0, \tag{1a}$$

$$\frac{\partial \rho u_i}{\partial t} + \frac{\partial}{\partial x_j}\left(\rho u_i u_j + p\,\delta_{ij} - \sigma_{ij}\right) = 0, \tag{1b}$$

$$\frac{\partial \rho E}{\partial t} + \frac{\partial}{\partial x_j}\left[(\rho E + p)u_j - \sigma_{ij}u_i + q_j\right] = 0. \tag{1c}$$

where $\rho$ is density, $u_i$ the velocity field, $p$ the static pressure, and $E = c_v T + \frac{1}{2}u_i u_i$ the specific total energy. $\delta_{ij}$ is the Kronecker delta. The conductive heat flux is $q_j = -\kappa\,\partial T/\partial x_j$, and the viscous stress tensor follows from Stokes' hypothesis (Stokes, 1845):

$$\sigma_{ij} = \mu(T)\left(\frac{\partial u_i}{\partial x_j} + \frac{\partial u_j}{\partial x_i} - \frac{2}{3}\frac{\partial u_k}{\partial x_k}\delta_{ij}\right). \tag{2}$$

The system is closed by the ideal gas equation of state $p = \rho R T$, with constant ratio of specific heats $\gamma = c_p/c_v = 1.4$. This calorically perfect gas assumption is admissible provided no vibrational excitation, chemical dissociation, or ionization of air constituents take place. These high-temperature non-equilibrium phenomena are therefore not modeled (Anderson, 2019). The aerodynamic heating still associated with hypersonic flight makes the temperature dependence of transport properties non-negligible; accordingly, the dynamic viscosity is evaluated via Sutherland's law (Sutherland, 1893),

$$\mu(T) = \mu_{\text{ref}}\left(\frac{T}{T_{\text{ref}}}\right)^{3/2}\frac{T_{\text{ref}} + S}{T + S}, \tag{3}$$

with reference values $\mu_{\text{ref}} = 1.716 \times 10^{-5}$ Pa s, $T_{\text{ref}} = 273.15$ K, and Sutherland constant $S = 110.4$ K. Thermal conductivity is recovered from $\kappa = \mu c_p/Pr$ with $Pr = 0.72$.

We highlight that the dataset generated for this application is built on steady-state flow field states of density, velocity, pressure and temperature. These are spatial fields defined on $(x, y, z)$ but not on time $t$.

## 4 CFD data generation

In this section we describe the CFD simulation data generation process, which is later used to train our artificial neural network models.

### 4.1 The Orion reentry capsule

The Orion reentry capsule was originally conceived by Lockheed Martin in the early 2000s for NASA's Constellation program, an attempt to return to the Moon before the year 2020, which was subsequently canceled. Orion was later redesigned for several other programs and is now expected to be used in NASA's Artemis program. The first unit, named

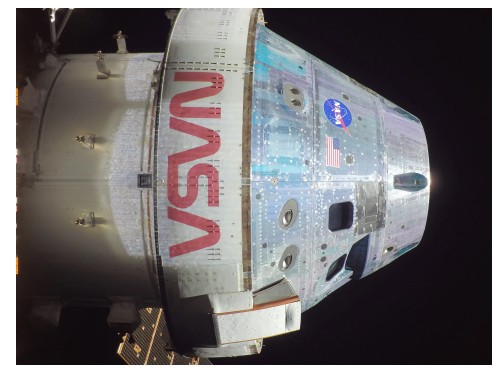

Figure 1: Orion, Artemis I, on flight day 13 (Nov. 28, 2022). Credit: NASA.

CM-002, was launched on November 16, 2022, on Artemis I, an uncrewed Moon-orbiting mission (see Figure 1). This project marked NASA's return to lunar exploration since the Apollo program five decades prior. In fact, in Greek mythology, Artemis is Apollo's twin sister, and NASA chose the name to explicitly echo the Apollo program while signaling a new era of lunar exploration. In particular, Orion provides protection from solar radiation, as well as advanced and reliable technologies for communication and life support, and, importantly for this work, enables high-speed entry into Earth's atmosphere (NASA, 2023).

### 4.2 Geometry

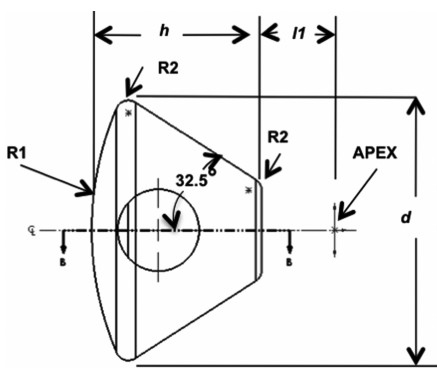

Figure 2: Orion CM Outer Mold Line. Figure from Brown et al. (2010).

While the geometry shown in Figure 1 incorporates numerous surface features such as windows, access hatches, and thruster ports, these details are neglected in the present work. The adopted geometry consists of a smooth, continuous outer mold line (OML) without local surface irregularities or protuberances. This idealized representation is appropriate for preliminary CFD simulations, where the primary objective is to capture the dominant aerodynamic behavior of the vehicle rather than the secondary effects associated with small-scale geometric features. More concretely, we utilize the OML geometry from the ballistic range tests reported in Brown et al. (2010): see Figure 2. The geometry is uniformly scaled to full scale using a maximum diameter of $d = 5$ m. The remaining geometric parameters are obtained by applying the same scaling to the reference dimensions.

### 4.3 Flow solver and models

The CFD solver STAR-CCM+ is used to perform steady 3D laminar simulations of the geometry. A third-order MUSCL central differencing (MUSCL-CD) scheme is employed for the spatial discretization of convective fluxes (Van Leer, 1979), and the inviscid fluxes are computed with the AUSM+ flux-vector splitting scheme (Liou, 1996). The simulations are performed at a Mach number $M = 5$ and an altitude of 50 kft. The static pressure and temperature are obtained from the International Standard Atmosphere (ISA) model, while the dynamic viscosity is computed using Sutherland's law. The freestream conditions are $T_0 = 1.30 \times 10^3$ K, $p_0 = 6.14 \times 10^6$ Pa, $M = 5.00$, and $Re_d = 9.68 \times 10^7$ (based on $d = 5$ m). Data is obtained for the geometry at these conditions and several AoAs in the range $0° \leq \alpha \leq 45°$. Simulations were not extended beyond $\alpha = 45°$, as the capsule is not expected to encounter such high AoAs during reentry under the flow conditions considered. Computations at higher angles were therefore deemed unnecessary.

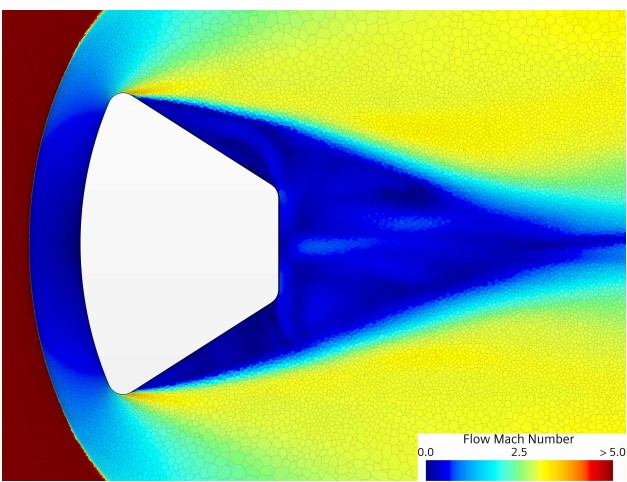

Figure 3: Converged 3D mesh of the Orion CM geometry at $M = 5$, $\alpha = 0°$. Centerplane cross-section overlaid with Mach number contour.

### 4.4 Simulation setup

The computational domain is a cylinder of diameter $10d$ and length $25d$. Freestream boundary conditions are applied to each boundary of the domain and the wall is defined as no-slip isothermal ($T_w = 300\,\text{K}$). The mesh is generated within STAR-CCM+ and consists of prism layer cells near viscous boundaries surrounded by polyhedral mesh elsewhere. A mesh convergence study with respect to aerodynamic coefficients of lift, drag, and pitch moment as well as with respect to the wall heat flux was first performed, leading to a final converged mesh of 22.8 million cells (130.7 million nodes) for the case at $\alpha = 0°$. Cases at high AoAs generally result in a lower number of cells due to the more slender bow-shock and smaller wake volume.

The final mesh is shown in Figure 3 for the case at $\alpha = 0°$: a centerplane cross-section of the mesh, superimposed on the computed Mach number field. The wake flow captured in the snapshot continues to oscillate despite residual convergence, as a steady simulation cannot fully converge an inherently unsteady wake (see Appendix A.1).

### 4.5 Angle-of-attack-dependent mesh adaptation

The flow around the Orion capsule experiences large modifications as the AoA $\alpha$ varies. Primarily, the change in orientation of the capsule facing the incoming stream causes the flow to rearrange itself, giving rise to different shock and expansion wave patterns and different wake topology. The adaptive mesh refinement strategy employed in the simulations allows the mesh to refine spatial regions where the flow experiences strong gradients, such as in the main bow shock ahead of the vehicle. Figure 4 shows the different meshes obtained at various incidences. In turn, this effectively means the 3D spatial coordinates $(x, y, z)$ are not fixed across all experiments but depend on AoA. These variables are taken into account by the model as explained thoroughly in Section 5.

### 4.6 CFD simulation time

Each simulation converged after approximately 60k iterations, equating to about 130 hours of wall time on 128 CPU cores, i.e., approximately 16.6k CPU-hours per simulation.

## 5 Model: 3D neural fields

Next, we describe our neural network design. The baseline model configuration takes as input 3D coordinates in space and AoA, and predicts static pressure, temperature, and the three components of velocity:

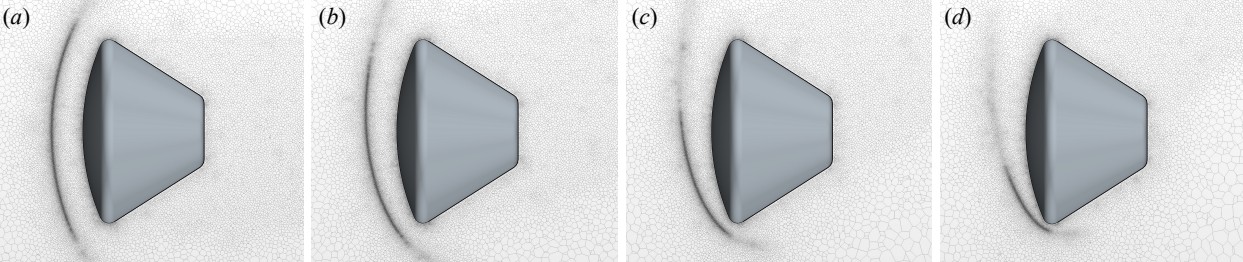

Figure 4: Computational mesh around the Orion capsule visualized on the centerplane $z = 0$ at (*a*) $\alpha = 0°$, (*b*) $\alpha = 15°$, (*c*) $\alpha = 30°$, (*d*) $\alpha = 45°$. Regions where the flow states experience strong gradients are displayed with denser cell distributions.

$$(p, T, v_x, v_y, v_z) = f_1(x, y, z, \alpha). \tag{4}$$

After training, the neural network can rapidly predict the flow field for any AoA. Also, unlike grid-based surrogates that require all training cases to be represented on a common discretization, our neural field is trained directly on physical coordinates sampled from independently adapted CFD meshes. As we have previously discussed, as the AoA changes, the shock location, wake structure, and mesh refinement pattern change as well (Figure 4); consequently, the training data is not defined on a fixed set of nodes and must also learn to generalize over heterogeneous meshes.

## 5.1 Fourier positional feature mappings

Naively fitting a multi-layer perceptron (MLP), as in Equation 4, leads to poor performance because MLPs struggle to capture high-frequency variations in the input given their spectral bias. These variations are particularly prevalent in high-speed aerodynamics due to the presence of shock waves (abrupt, nearly discontinuous changes in the properties of a fluid), which the neural network perceives as steep gradients in the training data. To address this issue, we leverage Fourier positional feature mappings (Tancik et al., 2020), which project the 3D coordinates, $\mathbf{x} = (x, y, z)$ to sinusoidal embeddings using a Gaussian projection matrix: $\Gamma(x, y, z) = \Gamma(\mathbf{x}) = [\cos(2\pi\mathbf{B}\mathbf{x}), \sin(2\pi\mathbf{B}\mathbf{x})]^T \in \mathbb{R}^{2m}$, where $\mathbf{B} \in \mathbb{R}^{m \times 3}$ is a random projection matrix sampled from $\mathcal{N}(0, \sigma^2)$. Both $m$ and $\sigma$ are hyperparameters which we adjust experimentally. Hence, the model with Fourier positional mappings can be described as:

$$(p, T, v_x, v_y, v_z) = f_2(\Gamma(x, y, z), \alpha). \tag{5}$$

Note that the Fourier feature mappings are applied only to the spatial coordinates. This design choice reflects the fact that, within the considered regime, the flow field varies "smoothly" with respect to AoA. Therefore, the inherent spectral bias of MLPs toward low-frequency functions is desirable, as it promotes smooth interpolation across angles and discourages spurious high-frequency variations along this global parameter. Equation 4 and Equation 5 constitute our "purely" data-driven baselines. Next we describe how we incorporate physical constraints into the model.

## 5.2 No-slip boundary condition

The no-slip boundary condition in fluid dynamics states that the fluid velocity at the wall is equal to the velocity of the wall itself. For a stationary wall, this means the fluid immediately next to the surface has zero velocity. Just above this layer of "stuck" fluid, the velocity gradually increases to match the free-stream flow. This gradual change in velocity forms a boundary layer: a thin region near the wall where viscous effects dominate. The boundary layer is critical in aerothermodynamics because it affects drag, surface heating, and the overall behavior of the flow. To model this effect in the neural network, we define $\kappa$ as the perpendicular

distance from the wall with $\kappa = 0$ on the surface (see Appendix A.2). We then scale the predicted velocity components to enforce the no-slip condition:

$$(p, T, v_x, v_y, v_z)_{\mathrm{BLv}} = \big(p, T, v_x(1 - e^{-\beta_1 \kappa}), v_y(1 - e^{-\beta_2 \kappa}), v_z(1 - e^{-\beta_3 \kappa})\big), \tag{6}$$

where $(p, T, v_x, v_y, v_z)$ are the outputs of the Fourier-based network $f_2$ and $\beta_1, \beta_2, \beta_3$ are learnable scalings. Importantly, $\kappa$ is not an input to the network; it is applied only to scale the velocity outputs after prediction. Pressure and temperature remain unchanged, so the rest of the model behaves identically to $f_2$. This approach biases the solution toward physically plausible boundary layer profiles while keeping the underlying neural network unchanged. Note that this trick does not simulate real viscous physics: the shape of the boundary layer is imposed by the function $1 - e^{-\beta \kappa}$, not by solving the Navier-Stokes equations. Also, in practice we are just modulating the network output near the wall rather than forcing a fixed exponential profile: the multiplicative factor ensures that the velocity goes to zero at the wall, but the neural network can still produce non-monotonic or complex variations in the boundary layer away from the wall. Thus, it is more flexible than a simple prescribed boundary layer.

### 5.3 Isothermal wall boundary condition

Specifying that a wall is isothermal means that the wall is maintained at a fixed temperature throughout the simulation, in our case this value is $T_w = 300$ K. This is often used to model realistic cases, such as actively cooled spacecraft surfaces, where engineers control the wall temperature to prevent overheating during high-speed flight. Importantly, an isothermal wall does not impose a particular temperature gradient at the surface; rather, the gradient develops naturally based on the flow conditions and heat transfer between the fluid and the wall. This boundary condition influences the behavior of the boundary layer, heat flux to the surface, and overall aerothermal characteristics, making it a useful approximation when the wall temperature can be actively regulated. Similar to the no-slip boundary condition for velocity, enforcing an isothermal wall produces a thermal boundary layer. To encourage the network to respect this condition, we scale the predicted temperature based on the perpendicular distance from the wall:

$$T_{\mathrm{BLT}} = T_w + \big(T - T_w\big)\big(1 - e^{-\beta_4 \kappa}\big), \quad \text{with } T = f_2^T(\Gamma(x, y, z), \alpha), \tag{7}$$

where $T$ is the temperature predicted by the Fourier-based network $f_2$ and $\beta_4$ is a learnable scaling. At the wall ($\kappa = 0$), this ensures $T_{\mathrm{BLT}} = T_w$, while away from the wall the network retains full flexibility to produce complex temperature profiles.

Note that the no-slip and isothermal-wall constraints bias the solution toward physically plausible near-wall behavior, but do not constitute a full PDE-constrained formulation.

## 6 Other surrogate alternatives

In this section, we motivate the use of neural fields as compared to other surrogate models. We perform an empirical exploration in Section 7.2.

### 6.1 Neural operators

It is useful to distinguish the learning problem addressed in this work from the one typically considered in the neural operator literature (Kovachki et al., 2021). In our particular case, the target quantity is a parametric flow field $\mathbf{q}_\alpha(\mathbf{x}) = \big(p(\mathbf{x}, \alpha), T(\mathbf{x}, \alpha), v_x(\mathbf{x}, \alpha), v_y(\mathbf{x}, \alpha), v_z(\mathbf{x}, \alpha)\big)$, $\quad \mathbf{x} = (x, y, z) \in \mathbb{R}^3$. Our model directly approximates this field through a coordinate-based representation $\hat{\mathbf{q}}_\alpha(\mathbf{x}) = f_\theta(\mathbf{x}, \alpha)$, or, with Fourier features as in Equation 5, $\hat{\mathbf{q}}_\alpha(\mathbf{x}) = f_\theta\big(\Gamma(\mathbf{x}), \alpha\big)$. This is a *neural field*: the spatial coordinate $\mathbf{x}$ is part of the input, and the network returns the predicted state at that location.

By contrast, a neural operator aims to learn a mapping between *functions*. In abstract form, one seeks an operator $\mathcal{G} : \mathcal{A} \to \mathcal{U}$, where $\mathcal{A}$ is a space of input functions or conditions and $\mathcal{U}$ is a space of output solution

fields. In the context of this work, one may idealize such a mapping as $\mathcal{G}(\alpha) = \mathbf{q}_\alpha(\cdot)$, that is, the input is the operating condition $\alpha$, while the output is the *entire* flow field $\mathbf{q}_\alpha(\cdot)$ over space. Evaluating the predicted solution at a point $\mathbf{x}$ then corresponds to

$$\hat{\mathbf{q}}_\alpha(\mathbf{x}) = \big[\mathcal{G}_\theta(\alpha)\big](\mathbf{x}). \tag{8}$$

The key distinction is that for neural fields the coordinate $\mathbf{x}$ is an input to the model, whereas in Equation 8 the model outputs a full spatial field and $\mathbf{x}$ appears only when that field is evaluated. In simple terms, in a neural field, we ask the network for one point at a time. In a neural operator, the network gives the whole field at once.

This difference has important implications for our data regime. A neural operator is trained across multiple realizations of the underlying parametric problem, $\{\alpha^{(i)}, \mathbf{q}_{\alpha^{(i)}}(\cdot)\}_{i=1}^N$, and therefore benefits primarily from having a large number $N$ of distinct CFD solutions. In contrast, the neural-field formulation exploits the fact that each single CFD realization provides a very large number of supervised spatial samples, $\{(\mathbf{x}_j, \alpha^{(i)}), \mathbf{q}_{\alpha^{(i)}}(\mathbf{x}_j)\}_{j=1}^{M_i}$, with $M_i$ equal to the number of mesh points in the simulation $i$. In our case, $N$ is small because each hypersonic CFD run is extremely expensive (130 hours each), while each $M_i$ is very large due to the high-resolution three-dimensional mesh. This makes the coordinate-based neural-field formulation particularly well suited to our setting, since it effectively transforms a "small data" problem into a "big data" problem and is hence strategic for this specific engineering bottleneck.

Many widely used neural operator architectures (Li et al., 2021) are most convenient on structured discretizations, whereas our CFD data is defined on large unstructured meshes with locally refined resolution near shocks and walls (Section 4). Coordinate-based neural fields avoid this restriction, since they only require point locations and the corresponding target states. For this reason, they provide a simple and natural way to learn continuous surrogates on irregular CFD discretizations.

We note, however, that this limitation applies primarily to classical FFT-based neural operators. Recent geometry-aware neural operators, such as GINO (Li et al., 2023), explicitly address irregular discretizations and complex three-dimensional geometries by combining graph-kernel operations, point-cloud or signed-distance-function representations, and latent structured grids. These methods are highly relevant to the broader problem of geometry-varying CFD surrogates. Our setting differs in two respects: first, we consider a fixed Orion capsule geometry with only a small number of extremely expensive hypersonic CFD realizations, rather than a large ensemble of varying geometries; second, the Mach-5 reentry simulations rely on STAR-CCM+ shock-adapted mesh refinement, whose final mesh connectivity is not readily exportable for mesh-based operator baselines. In contrast, existing GINO benchmarks use more standard OpenFOAM-based vehicle-aerodynamics settings, such as Ahmed-body and car geometries, where large geometry sweeps and mesh access are substantially more feasible. A full comparison with geometry-informed neural operators on hypersonic, geometry-varying capsule datasets is therefore an important direction for future work.

## 6.2 Graph neural networks

Following this discussion, a natural alternative is GNNs, which are able to operate on irregular grids. The first challenge in this case is that the standard commercial software used for large-scale CFD simulations with advanced mesh refinement does not typically allow one to export the final mesh. As a consequence, we cannot do message passing based on the simulation grid, and we must construct k-NN graphs using spatial proximity instead. In Figure 5, we visualize 2D slices of some of the 3D k-NN graphs used in our experiments.

A message passing GNN layer $l$ (ignoring edge and graph level features for simplicity here) is computed as

$$\mathbf{h}_i^{(l+1)} = \phi\Big(\mathbf{h}_i^{(l)}, \bigoplus_{j \in \mathcal{N}(v_i)} \psi(\mathbf{h}_i^{(l)}, \mathbf{h}_j^{(l)})\Big). \tag{9}$$

Note that here we use $i$ and $j$ to index (latent) features referring to the node $v_i$ (not to be confused with velocity) at which we perform a computation and its neighboring node, respectively. $\mathcal{N}(v_i)$ is the one-hop neighborhood. Also, $\psi$ is a message passing function, $\bigoplus$ is an aggregation function, which must be

permutation-invariant, and lastly, $\phi$ is a readout or update function. Depending on the GNN, $\psi$ and $\phi$ are implemented differently. Importantly, the update equation is local and it is only dependent on the one-hop neighborhood of the node at each layer. For clarity, Equation 9 can be further decoupled into three update rules (Borde & Bronstein, 2025):

$$\mathbf{m}_{ij}^{(l)} \leftarrow \psi(\mathbf{h}_i^{(l)}, \mathbf{h}_j^{(l)}); \quad \mathbf{a}_i^{(l)} \leftarrow \bigoplus_{j \in \mathcal{N}(v_i)} \mathbf{m}_{ij}^{(l)}; \quad \mathbf{h}_i^{(l+1)} \leftarrow \phi(\mathbf{h}_i^{(l)}, \mathbf{a}_i^{(l)}) \qquad \text{(Message, Aggregate, Update)}$$

This localized inductive bias is beneficial for many classical graph representation learning tasks, such as transductive learning on citation networks (Yang et al., 2016), where neighboring nodes often have similar labels or features. However, it can also become a weakness in hypersonic flow prediction, as we show in Section 7.2. Standard message-passing GNNs tend to exhibit a smoothing bias (Kipf & Welling, 2017): repeated neighborhood aggregation mixes information across nearby nodes and can therefore behave like a low-pass filter. This is counterproductive when the target fields contain sharp gradients and near-discontinuities, such as shocks and boundary layers in hypersonic flows.

## 7 Quantitative results and aerothermodynamic analysis

In the following section we empirically evaluate our methodology. We start by studying the importance of Fourier mappings in the context of hypersonic flow prediction, in particular focusing on our 3D generalization of previous 2D studies (Borde et al., 2023). Next, we perform a one-to-one comparison with other surrogate alternatives from Section 6, which we find to be suboptimal for hypersonics. Lastly, we discuss the importance of enforcing boundary conditions, and analyze the final simulation time as compared to traditional CFD.

### 7.1 On the importance of Fourier mappings for learning hypersonic flow features

The neural field architecture is trained using a 10-layer MLP with a hidden dimension of 1024. We utilize the Adam optimizer and MSE loss for 20 epochs with a learning rate of $10^{-4}$ and a mini-batch size of

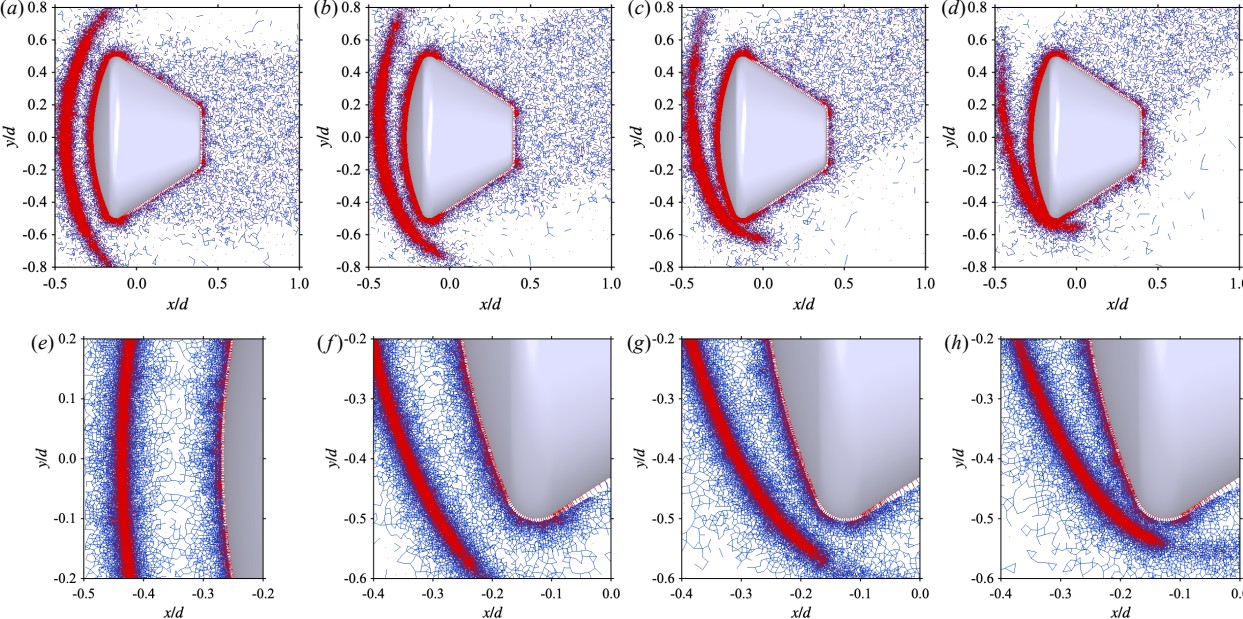

Figure 5: k-NN graphs ($k = 6$) extracted within $|z| < 0.02/d$. Full capsule view and close-ups in the shock layer region for $(a, e)$ $\alpha = 0°$, $(b, f)$ $\alpha = 15°$, $(c, g)$ $\alpha = 30°$, $(d, h)$ $\alpha = 45°$. Mesh points (red markers), k-NN edges (blue lines).

256. We start our experimental procedure by calibrating the model hyperparameters. All learnable scaling coefficients are initialized to $\beta_1 = \beta_2 = \beta_3 = \beta_4 = 5$, but are free to vary independently during optimization. The AoA training split consists of $\{0°, 5°, 10°, 20°, 25°, 35°, 40°, 45°\}$ while validation is performed on the unseen angles $\{15°, 30°\}$.

In particular, we compare the baseline configuration presented in Equation 4 to that in Equation 5. In Table 1, we observe that adding Fourier mappings improves performance as compared to passing the coordinates directly to the MLP, as expected. We perform a grid search over $\sigma \in \{5, 15, 30, 45, 60, 75, 85, 90, 100\}$. The parameter $\sigma$ directly controls the frequency spectrum of functions the network can represent and learn efficiently. For low values of $\sigma \in [5, 15]$, the network behaves similarly to a standard MLP with a strong spectral bias toward low frequencies, and hence promotes learning smooth, slowly varying functions. This leads to underfitting near shocks and smeared discontinuities. On the other hand, for high values of $\sigma > 75$, very high-frequency components dominate, and the network can represent extremely sharp features, but optimization becomes harder and there is a risk of fitting noise or mesh artifacts. Indeed, we find the sweet spot to be at intermediate values, which provide a balanced spectrum and allow the network to effectively learn both smooth free-stream flow features and sharp gradients. Concretely, $\sigma = 45$ empirically provides a good trade-off in terms of expressivity and generalization. We use this configuration for all experiments in the next subsection.

Table 1: Final validation MSE for different encoding configurations.

| Encoding | None (Base model) | $\sigma$=5 | $\sigma$=15 | $\sigma$=30 | $\sigma$=45 | $\sigma$=60 | $\sigma$=75 | $\sigma$=85 | $\sigma$=90 | $\sigma$=100 |
|---|---|---|---|---|---|---|---|---|---|---|
| Val. MSE | 0.01539 | 0.00856 | 0.00726 | 0.00635 | **0.00484** | 0.00485 | 0.00526 | 0.00528 | 0.00536 | 0.00557 |

## 7.2 Empirical comparison with other surrogate alternatives

Following the discussion in Section 6, we focus on comparing neural fields to GNN-based surrogates. We consider different types of GNNs: GCN (Kipf & Welling, 2017), GAT (Veličković et al., 2018), GATv2 (Brody et al., 2022), GIN (Xu et al., 2019), ChebNet (Defferrard et al., 2016), GraphGPS (Rampášek et al., 2023), MeshGraphNet (Pfaff et al., 2021), Graph U-Net (Gao & Ji, 2019) and SGC (Wu et al., 2019). Each model uses 10 layers and is augmented with Fourier positional feature mappings with $\sigma = 45$ (based on our best configuration in Table 1). This layer count is chosen to roughly match the number of parameters of the MLP-based neural field (approximately 20 million trainable weights and biases).

We construct k-NN graphs using the 3D spatial coordinates of the nodes (Figure 5). To investigate the impact of graph connectivity, we perform an ablation study by varying the parameter $k$, which determines the number of neighbors per node. Note that the effective receptive field (depth) of the GNN corresponds to its number of layers. GraphGPS differs from purely message-passing architectures because its global attention mechanism allows each node to aggregate information from all other nodes in the subsampled graph within a layer. However, this does not increase the message-passing depth itself, and for computational tractability both message passing and attention are performed on the same subsampled graph of 5000 nodes. As we can see in Table 2, as we increase $k$ the validation loss increases. $k = 2$ (the sparsest case we explore) gives the best performance, although this is still worse than the neural field with the same frequency encoding, which achieves a validation loss of 0.00484.

In Table 3, we further average results over 3 random seeds for $k = 2$, and test removing the Fourier mappings. We can see that Fourier mappings help GNN performance too. The GAT model performs slightly better than GCN and SGC, but it still does not outperform the neural field. Stronger graph architectures, including GIN, ChebNet, Graph U-Net, and MeshGraphNet, achieve lower validation losses, with ChebNet and Graph U-Net performing best among the GNNs considered. However, none surpass the neural field baseline. GraphGPS, despite incorporating global attention and not being constrained by the k-NN graph, achieves comparable performance to GAT and remains substantially worse than the neural field. This empirically supports our discussion in Section 6.2 regarding the unsuitability of classical GNNs for hypersonic flow modeling, as well as some of their stronger variants. Even when incorporating more expressive message-passing operators or global attention, these graph-based models remain substantially worse than the coordinate-based neural field.

Table 2: Validation loss for GNNs with different $k$ values and $\sigma = 45$.

| $k$ | 2 | 4 | 6 | 8 | 10 |
|---|---|---|---|---|---|
| GCN | 0.00710 | 0.00714 | 0.00782 | 0.00947 | 0.00923 |
| GAT | 0.00663 | 0.00670 | 0.00678 | 0.00775 | 0.00815 |
| GIN | 0.00573 | 0.00588 | 0.00604 | 0.00671 | 0.00712 |
| ChebNet | 0.00501 | 0.00507 | 0.00524 | 0.00596 | 0.00642 |
| SGC | 0.00748 | 0.00762 | 0.00805 | 0.00911 | 0.00954 |
| Graph U-Net | 0.00513 | 0.00527 | 0.00611 | 0.00664 | 0.00692 |
| MeshGraphNet | 0.00551 | 0.00537 | 0.00562 | 0.00615 | 0.00648 |
| GraphGPS | 0.00662 | 0.00681 | 0.00734 | 0.00750 | 0.00780 |

This is further supported by studying the dissipative behavior of GNNs at the hypersonic bow shock and the boundary layer in Section 8.1. For completeness, we also include classical machine-learning baselines, such as linear regression, ridge regression, a k-nearest neighbors regressor, and random forests in Table 3.

Table 3: Validation loss (mean $\pm$ std over 3 seeds) for GCN, GAT, and other baselines.

| Model | No Fourier | Fourier ($\sigma = 45$) |
|---|---|---|
| GCN | $0.06771 \pm 0.00093$ | $0.00706 \pm 0.00053$ |
| GAT | $0.06645 \pm 0.00311$ | $0.00682 \pm 0.00068$ |
| GIN | $0.06888 \pm 0.00036$ | $0.00570 \pm 0.00010$ |
| ChebNet | $0.06592 \pm 0.00102$ | $0.00511 \pm 0.00031$ |
| SGC | $0.06984 \pm 0.00127$ | $0.00752 \pm 0.00059$ |
| Linear | $0.07380 \pm 0.00240$ | $0.01490 \pm 0.00110$ |
| Ridge | $0.07160 \pm 0.00190$ | $0.01280 \pm 0.00095$ |
| KN Regressor | $0.06940 \pm 0.00210$ | $0.01110 \pm 0.00074$ |
| Random Forest | $0.06620 \pm 0.00150$ | $0.00985 \pm 0.00051$ |
| Graph U-Net | $0.048213 \pm 0.0019$ | $0.00534 \pm 0.00027$ |
| MeshGraphNet | $0.044626 \pm 0.0015$ | $0.00553 \pm 0.00022$ |
| GraphGPS | $0.052884 \pm 0.0021$ | $0.00667 \pm 0.00031$ |

Lastly, it is important to note that we cannot further test message-passing propagation over the original mesh connectivity structure, since most commercial CFD software does not allow exporting it. This is a further logistical constraint for GNNs. Nevertheless, by visually inspecting Figure 3, we can observe that most nodes have 4 or more neighbors and are also connected to nearby nodes in terms of spatial coordinates (when looking at the 2D slice). Ergo, we would not expect a GNN that uses the original mesh adjacency matrix to outperform the neural fields; instead, its performance is likely to be similar to the results in Table 2. Although the exact performance discrepancy is hard to quantify, both the graph derived from the mesh and the k-NN graphs share the same locality inductive bias.

### 7.3 Adding physical boundary conditions

To assess the impact of the boundary-condition enforcement, we compare four configurations: no boundary conditions, no-slip velocity ($BC_v$), isothermal wall ($BC_T$), and both combined. As shown in Table 4, enforcing boundary conditions substantially improves performance in unseen AoAs, with the largest gains achieved when both velocity and thermal wall conditions are applied. We use the best-performing model for all subsequent analyses.

Table 4: Effect of boundary condition enforcement on val AoAs (mean $\pm$ std across runs, 3 seeds). $\sigma = 45$.

| Boundary conditions | None | $BC_v$ | $BC_T$ | $BC_v + BC_T$ |
|---|---|---|---|---|
| MLP | $0.00483 \pm 0.00039$ | $0.00444 \pm 0.00035$ | $0.00292 \pm 0.00181$ | $\mathbf{0.00228 \pm 0.00002}$ |

We note that additional preliminary experiments with GATv2 show that it is competitive when implemented without boundary conditions, but its performance does not match that of neural fields when combined with the no-slip velocity and isothermal-wall temperature constraints (achieving an MSE of $0.00274 \pm 0.00009$ vs $0.00228 \pm 0.00002$ in the case of the neural field).

### 7.4 Additional generalization experiments: interpolation and extrapolation across angle of attack and coarse to fine meshes

To further test the robustness of the proposed model, we also conduct additional experiments to evaluate generalization across both AoA regimes and spatial mesh resolution. For AoA, we consider three train/test splits that distinguish extrapolation from sparse interpolation. Here all models use both boundary conditions. High-AoA extrapolation trains on $\{0°, 5°, 10°, 15°, 20°, 25°, 30°, 35°\}$ and tests on higher unseen angles $\{40°, 45°\}$. Low-AoA extrapolation trains on $\{10°, 15°, 20°, 25°, 30°, 35°, 40°, 45°\}$ and tests on lower unseen angles $\{0°, 5°\}$. Sparse-AoA interpolation trains only on the endpoint angles $\{0°, 5°, 40°, 45°\}$ and tests on the intermediate angles $\{10°, 15°, 20°, 25°, 30°, 35°\}$. As shown in Table 5, sparse interpolation gives the lowest error, while high- and low-AoA extrapolation yield slightly higher errors. These results indicate that the model generalizes reasonably across unseen AoAs, although extrapolation outside the training range remains more challenging than interpolation within it, as expected. See Appendix B.2 for visualizations.

We also test coarse-to-fine spatial generalization by training across all AoAs on a coarsened version of each mesh and validating on the remaining finer-grained spatial points. In the 80/20 split, the model is trained on 80% of the points from each AoA case and evaluated on the held-out 20%; in the 50/50 split, only half of the spatial points are used for training. This evaluates whether the coordinate-based surrogate can infer flow quantities at finer spatial locations that are not directly observed during training. As shown in Table 6, the model achieves a best validation MSE of $0.00270 \pm 0.00041$ for the 80/20 split and $0.00487 \pm 0.00051$ for the more challenging 50/50 split. The increase in error as fewer spatial points are used for training reflects the expected difficulty of extrapolating from coarser to finer spatial supervision, while still retaining good accuracy on unseen mesh points.

Table 5: Generalization across AoA regimes using the $BC_v + BC_T$ MLP model. Results reported as best validation MSE (mean $\pm$ std across seeds).

| Setting | Best Val. MSE |
|---|---|
| High-AoA Extrapolation | $0.00621 \pm 0.00027$ |
| Low-AoA Extrapolation | $0.00720 \pm 0.00031$ |
| Sparse Interpolation | $0.00554 \pm 0.00018$ |

Table 6: Generalization across unseen spatial points using the $BC_v + BC_T$ MLP model. Results reported as best validation MSE (mean $\pm$ std across seeds).

| Train/Test Split | Best Val. MSE |
|---|---|
| 80% train points / 20% test points | $0.00270 \pm 0.00041$ |
| 50% train points / 50% test points | $0.00487 \pm 0.00051$ |

### 7.5 Neural inference time

Each full-field prediction (pressure, temperature, and all velocity components) takes less than 5 seconds on a single node with 8 H100 GPUs, parallelized with a batch size of 2,097,152 per GPU, or approximately 37 seconds when run serially on a single H100. Training each model takes approximately 50 minutes per node. Prediction time can be reduced further through increased parallelization (e.g., to under 3 seconds using two nodes) or using optimized CUDA kernels, which we do not explore here. This contrasts with 130 hours of wall time using traditional CFD (Section 4). That is, a $93,600\times$ speedup when using an H100 node, and a $156,000\times$ when using two.

## 8 Aerothermodynamic analysis

Next, we evaluate the model's predictions from an aerodynamic perspective.

### 8.1 Challenges capturing sharp discontinuities in hypersonic flows

In Figure 6 we compare the error near the capsule (with respect to the CFD simulation) for the baseline neural field vs that produced by the neural field with Fourier $\sigma = 45$. More concretely, the mismatch between the ground truth and the model approximation is plotted using the normalized percentage error $\mathcal{E}_i = 100 \cdot \frac{|q_i - \tilde{q}_i|}{\max\left(|\tilde{q}_i| + |q_i|, \, \tau\right)}$ [%] where $q_i$ is the true state and $\tilde{q}_i$ the approximated state and $\tau$ is a small positive threshold used in the denominator so the percent error remains bounded when the states are near zero.

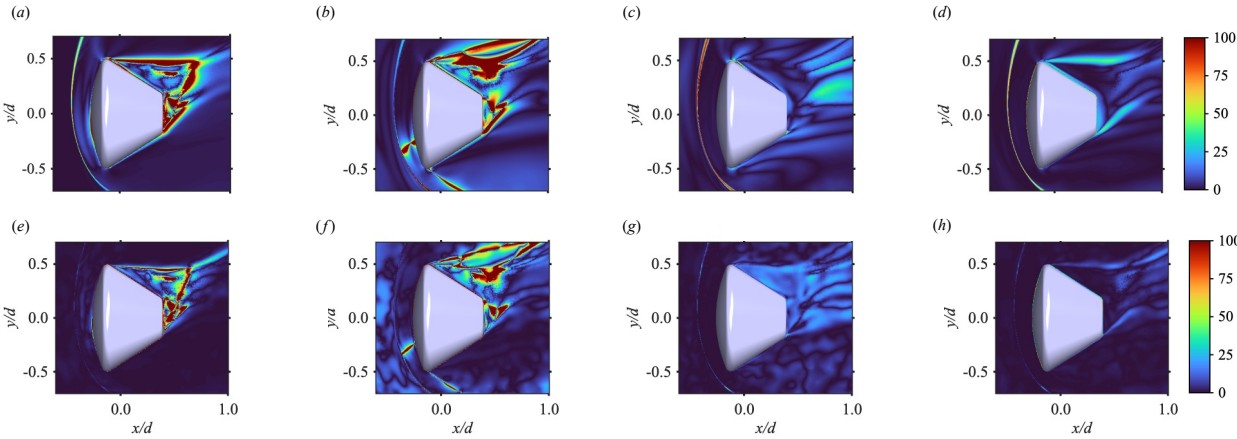

Figure 6: Flow field prediction error with respect to CFD ground truth on the $z/d = 0$ plane at $\alpha = 15°$ (validation AoA), comparing neural field models with $(e, f, g, h)$ and without Fourier features $(a, b, c, d)$, as in Table 1. In terms of columns, we have the horizontal $v_x/U_\infty$ $(a, e)$, vertical velocity $v_y/U_\infty$ $(b, f)$, pressure $p/p_\infty$ $(c, g)$, and temperature $T/T_\infty$ $(d, h)$. Contours show the normalized percentage error.

As shown, without Fourier positional feature mappings (top row), errors are localized around the shock and other features such as the free-shear layer, the Prandtl-Meyer expansions, and the recirculating flow behind the capsule (features critical to hypersonic flows), whereas with $\sigma = 45$ (bottom row), the error distribution becomes more uniform and practically disappears at the bow shock.

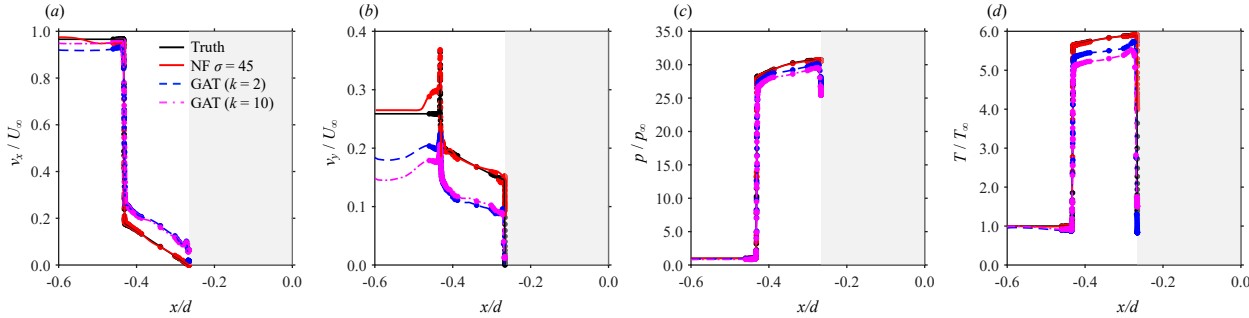

Figure 7: Shock gradient predictions along the $y/d = 0$ symmetry line and plane $z/d = 0$ for $\alpha = 15°$. (a) $v_x/U_\infty$, (b) $v_y/U_\infty$, (c) $p/p_\infty$, (d) $T/T_\infty$. The shaded gray area marks the capsule body.

Next, we show that GNNs struggle to capture hypersonic discontinuities. In particular, we compare the neural field model against GAT predictions with $k = 2$ and $k = 10$; all of them use Fourier positional feature

mappings with $\sigma = 45$ (Table 2). In Figure 7 we extract the main bow shock and boundary layer gradients by seeding the volumetric flow data along the symmetry line $y/d = 0$ on the symmetry plane $z/d = 0$, as conveniently done in aerodynamic 3D flows around symmetric bodies. In the present case, the free-stream velocity vector has projections only in $x$ and $y$ (since the sideslip angle is zero), therefore the flow has strong in-plane gradients that can be extracted from $z/d = 0$.

Additionally, to extract gradients in the $v_z$ velocity component, which otherwise remains virtually zero at $z/d = 0$ due to symmetry, the $y/d = 0$ line is sampled on the $z/d = 0.2$ plane, as illustrated in Figure 8. The GNNs (both with $k = 2$ and $k = 10$) clearly struggle to capture sudden jumps in velocity, pressure, and temperature near the shock. Notably, a more pronounced smoothing of the steep gradients in the flow states can be observed with a larger $k$, as expected due to the dissipative nature driven by neighborhood aggregation in message-passing. This aligns with the higher validation error in Table 2.

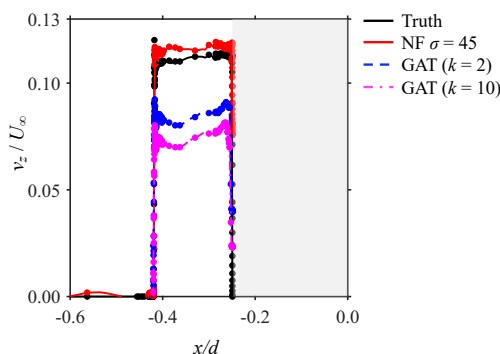

Figure 8: The normalized velocity $v_z/U_\infty$ is extracted on the $z/d = 0.2$ plane and along the $y/d = 0$ symmetry line ($\alpha = 15°$).

## 8.2 Flow field prediction

After determining that the neural field with Fourier positional feature mappings performs best (both from a machine learning metrics perspective and from an aerothermodynamic analysis perspective), we further explore the predictions of this configuration. In particular, we focus on the best model in Table 4, which uses all boundary conditions. The horizontal and vertical velocities $v_x$ and $v_y$, static pressure $p$ and static temperature $T$ spatial fields are displayed in Figure 9 for both the CFD data serving as the ground truth and the model prediction at $\alpha = 15°$. The data is extracted from a plane at $z/d = 0$ and interpolated on a structured $x$-$y$ grid with uniform spacing $\Delta x/d = 0.004$ and $\Delta y/d = 0.003$.

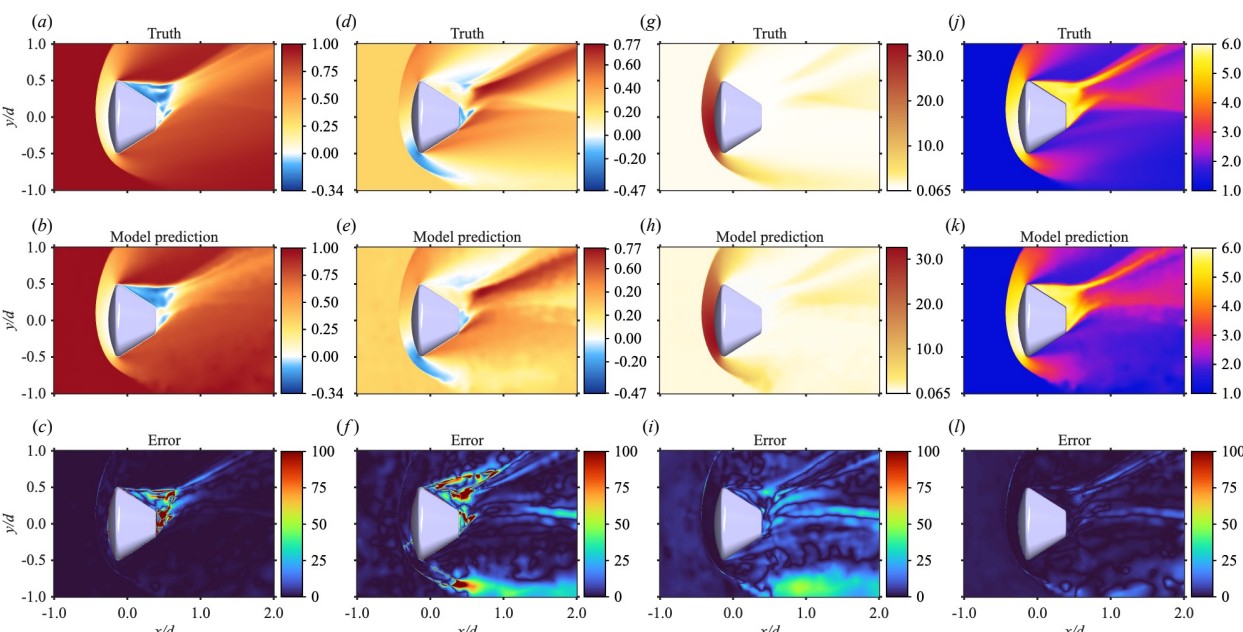

Figure 9: Comparison of the horizontal $v_x/U_\infty$ (a, b), vertical velocity $v_y/U_\infty$ (d, e), pressure $p/p_\infty$ (g, h), and temperature $T/T_\infty$ (j, k) fields between the ground truth and model predictions at $\alpha = 15°$. The normalized percentage error is shown in (c, f, i, l). Contours extracted on the plane $z/d = 0$.

The (*a, b*) panels in Figure 9 show that the model has learned the main flow features. The bow shock is resolved exceptionally well and also the dominant wake orientation and separation region are captured. Most of the error (panel *c*) is within the recirculation zone; however, the fine details of the flow therein are not physical given the unsteady nature of the flow and the type of simulations employed (see Appendix A.1). Any inaccuracies from the model are therefore not concerning. Similar observations can be drawn from the vertical velocity $v_y$ (panels *d, e, f*) except for the lower side of the bow shock in $x/d \in [0.0, 1.0]$, $y/d \in [-1.0, -0.5]$, where the model prediction displays some noise. This is likely due to the lower-density mesh in this region of the flow. Note that the lower bow shock region of the flow is of limited engineering interest as it does not influence the pressure or heat flux on the vehicle, justifying the lower mesh density. For the sake of clarity, these inaccuracies were found also in the non-interpolated data, so they are not a mere visualization issue but a real model behavior. The pressure and temperature states (panels *g, h, i* , and *j, k, l*) are reconstructed very accurately; the pressure being affected in the lower bow shock area seemingly for a similar reason to the vertical velocity.

The overall $\mathcal{L}_2$-norm based error is computed from the original CFD grid points as $\mathcal{L}_2 = 100 \cdot \frac{\|q_i - \tilde{q}_i\|_2}{\|q_i\|_2 + \|\tilde{q}_i\|_2}$ [%]. The $\mathcal{L}_2$ errors are 4.92% for $v_x$, 8.20% for $v_y$, 11.97% for $v_z$, 3.04% for pressure $p$, and 6.24% for temperature $T$. All states except the $v_z$ velocity (not shown for brevity in Figure 9) are approximated with less than 10% error, with the pressure state $p$ displaying only $\approx 3\%$ error. The $v_z$ velocity is the least accurate, perhaps due to its comparably smaller magnitude.

## 8.3 Pressure distribution prediction on the surface of the Orion reentry capsule

In high-speed bluff body flows, the major contributor to the total aerodynamic drag exerted on the vehicle originates from pressure forces (Anderson, 2003).

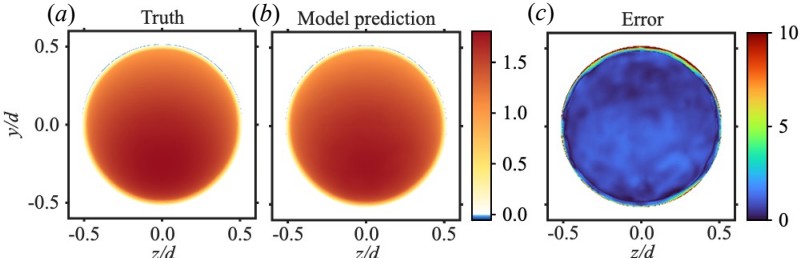

Figure 10: Comparison of the pressure coefficient $C_P$ on the frontal surface of the capsule for (*a*) the ground truth and (*b*) the model prediction at $\alpha = 15°$. The normalized percentage error is presented in (*c*).

Hence, we examine the predictive accuracy of the model on the pressure coefficient distribution. Figure 10 presents a comparison of the pressure coefficient $C_P$, computed as

$$C_P = \frac{\frac{p}{p_\infty} - 1}{0.5 \gamma M_\infty^2}. \tag{10}$$

The figure reveals that the model predicts the distribution over the capsule forebody to a high degree of accuracy, with maximum error within 10% occurring on the shoulder (Figure 11) where the flow accelerates through an expansion fan. Strong pressure gradients are therefore present in this region of flow and prediction is clearly more challenging compared to locations with more gentle surface curvature. Nonetheless, the recorded error is acceptable.

These results indicate that the pressure coefficient derived from the predicted pressure field is successfully reconstructed for an AoA not seen by the model during the training phase. The same set of plots presented in this section is repeated in Appendix B for $\alpha = 30°$.

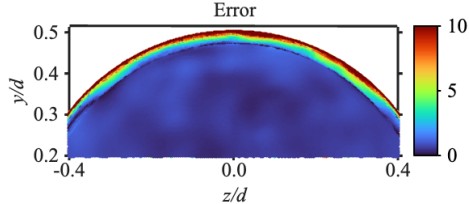

Figure 11: Zoomed-in view of the error on the $C_P$ prediction over the capsule shoulder region at $\alpha = 15°$.

## 9 Conclusion

We have shown that neural fields can serve as effective hypersonic surrogate models, with the potential to reduce simulation time from hundreds of hours to seconds. Although CFD can also be parallelized, the iterative nature of its solvers often limits strong scaling, so adding nodes typically does not yield near-linear reductions in wall time; in contrast, batched neural inference does, which is another key strength of our method. Our results also corroborate that Fourier features can effectively learn sharp discontinuities characteristic of hypersonic flows and that enforcing physics-informed boundary conditions substantially improves model performance. Finally, we have characterized the limitations of alternative surrogate models, concluding that neural fields provide superior logistical flexibility by bypassing the high data requirements of neural operators and the mesh-connectivity constraints of GNNs, while simultaneously capturing the complex discontinuities inherent to the hypersonic regime.

This work is primarily an application-focused empirical study and dataset contribution for 3D hypersonic aerothermodynamic surrogates. Our overarching long-term objective is to streamline the processes of aero-dynamic design, mission analysis, and control, enabling faster and more thorough exploration of potential solutions during the initial phases of space mission development. Achieving this vision requires general models capable of effectively simulating the flow field around any type of geometry, which is, of course, a data-hungry endeavor.

## Acknowledgments

Pietro Innocenzi acknowledges the Department of Mechanical Engineering at Imperial College London for providing access to the STAR-CCM+ license and high-performance computing facilities used in this research.

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

# A    Additional CFD data generation details

In this appendix we provide additional information regarding the CFD simulations and the calculation of other variables used for training the models.

## A.1    Wake oscillations

A steady CFD simulation cannot converge an unsteady wake. Consequently, even when all other flow quantities are converged, the residuals associated with the wake oscillate between low values, and the flow field in the wake region changes from one iteration to another (see Figure 12).

Generally, this issue is addressed by computing a time-averaged solution of the entire flow field (in steady simulations, an *iteration-averaged* solution), in which all flow variables are averaged over a number of time steps (or iterations) corresponding to several convective time scales. This approach was not adopted in the present simulations but will be considered in future work. Instead, the neural network was trained using a single snapshot of the flow field at each AoA.

While this choice does not affect the shocks or boundary layers on the front of the model, which are essentially steady, it does influence the wake, which remains inherently unsteady and varies between iterations. As a consequence, larger prediction errors from the neural network are expected in this region of the flow.

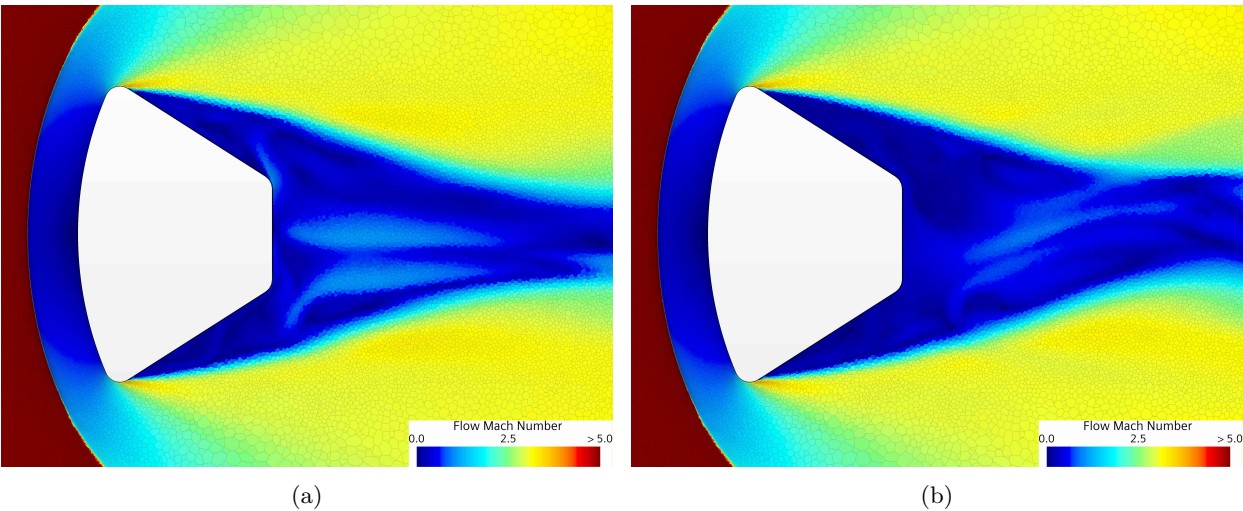

(a)                                                                                   (b)

Figure 12: Snapshot Mach number contour over the centerplane cross-section at $M = 5$, $\alpha = 0°$. The Mach number values in the wake are different between the two iterations because of the unsteadiness in this region of the flow.

## A.2    Wall Distance Computation

Let $\mathcal{S} \subset \mathbb{R}^3$ denote the set of points on the capsule surface. For any point $\mathbf{x} = (x, y, z) \in \mathbb{R}^3$, the wall distance $\kappa$ is defined as:

$$\kappa(\mathbf{x}) = \min_{\mathbf{s} \in \mathcal{S}} \|\mathbf{x} - \mathbf{s}\|_2. \tag{11}$$

To efficiently evaluate $\kappa$, a k-d tree is constructed from the wall point cloud using SciPy's `cKDTree` class. For each computational point $x$, the wall distance is obtained by querying the tree for the nearest surface point. Points identified as wall locations are explicitly assigned the value $\kappa = 0$, while for all other points $\kappa$ equals the Euclidean distance to the nearest wall point returned by the k-d tree. The computation is performed in chunks to allow processing of the full dataset without exceeding memory limits, and the resulting values of the $\kappa$ field are stored for subsequent analysis.

# B  Additional aerothermodynamic analysis results and visualizations

In this appendix, we present extended visualizations that complement the discussion in Section 7.

## B.1  Flow prediction for interpolation results

In Figures 13 and 14 we show the prediction at $\alpha = 30°$ and report $\mathcal{L}_2$ errors of 5.09% for $v_x$, 6.55% for $v_y$, 11.50% for $v_z$, 1.58% for pressure $p$, and 6.19% for temperature $T$. Similarly to the $\alpha = 15°$ incidence case, the shock and wake structures are well approximated by the model as well as the pressure distribution on the frontal surface of the vehicle.

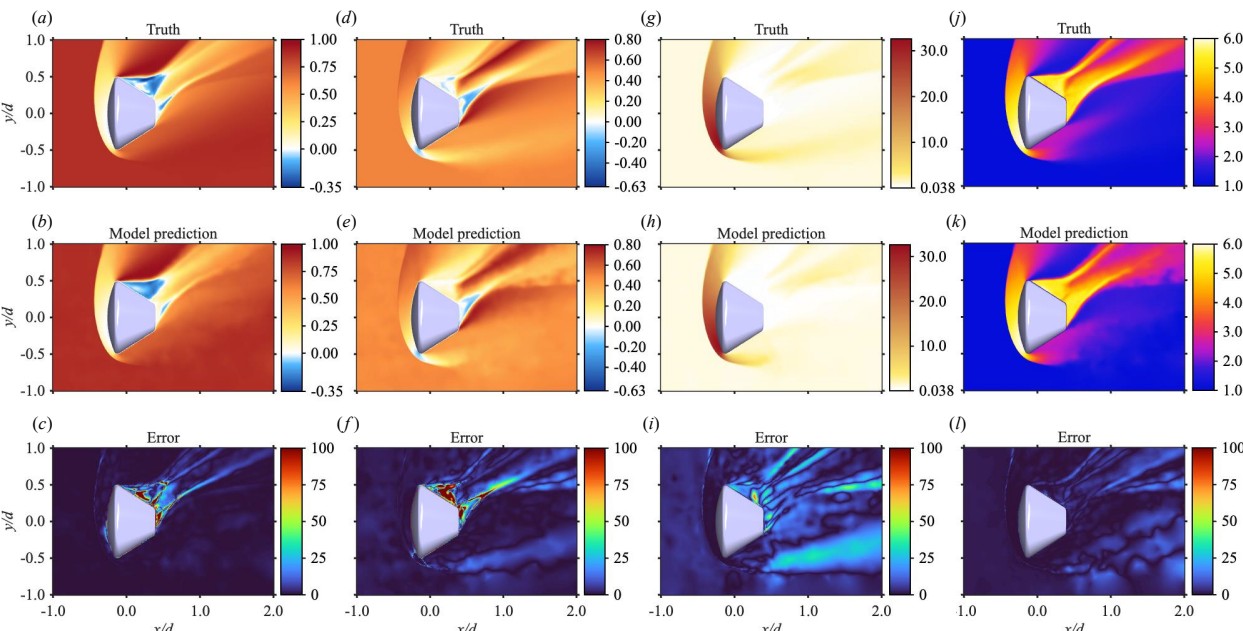

Figure 13: Comparison of ground truth and (best) model predictions at $\alpha = 30°$ for: horizontal velocity $v_x/U_\infty$ (a, b), vertical velocity $v_y/U_\infty$ (d, e), pressure $p/p_\infty$ (g, h), and temperature $T/T_\infty$ (j, k). The normalized percentage error is shown in (c, f, i, l). All contours are extracted on the plane $z/d = 0$.

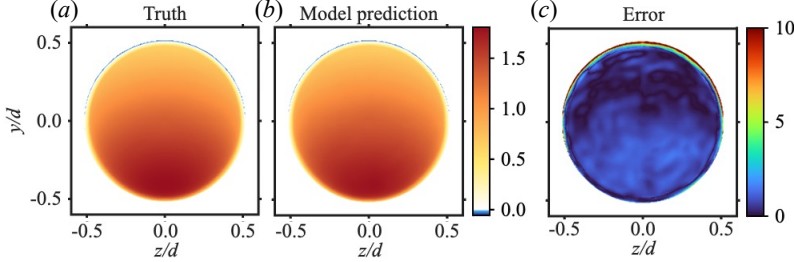

Figure 14: Comparison of the pressure coefficient $C_P$ on the frontal surface of the capsule for (a) the ground truth and (b) the model prediction at $\alpha = 30°$. The normalized percentage error is presented in (c).

## B.2  Flow prediction for extrapolation results

Figures 15–18 show the flow field predictions for the four extrapolation cases $\alpha \in \{0°, 5°, 40°, 45°\}$. The $\mathcal{L}_2$ errors for $v_x$, $v_y$, $v_z$, $p$, and $T$ are: 7.35%, 13.56%, 18.50%, 3.90%, 13.91% at $\alpha = 0°$; 7.85%, 15.17%, 19.97%, 3.95%, 15.38% at $\alpha = 5°$; 8.73%, 10.45%, 20.09%, 4.07%, 10.64% at $\alpha = 40°$; and 8.94%, 10.92%, 20.59%, 4.03%, 11.11% at $\alpha = 45°$.

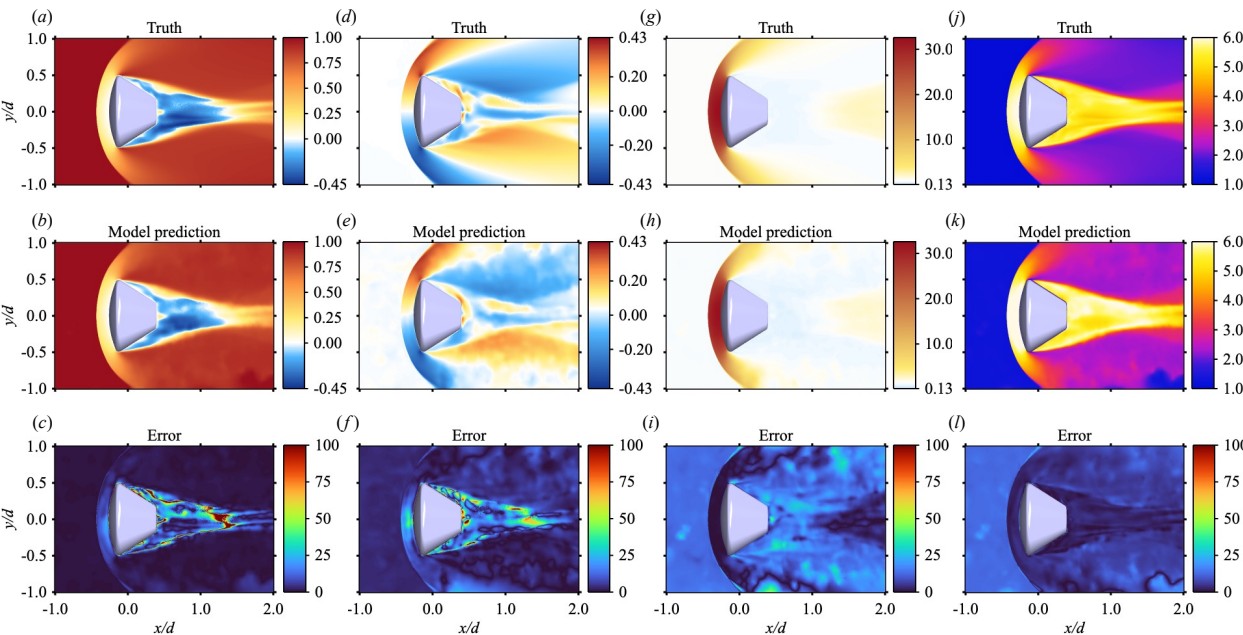

Figure 15: Comparison of ground truth and (best) model predictions at $\alpha = 0°$ for: horizontal velocity $v_x/U_\infty$ (a, b), vertical velocity $v_y/U_\infty$ (d, e), pressure $p/p_\infty$ (g, h), and temperature $T/T_\infty$ (j, k). The normalized percentage error is shown in (c, f, i, l). All contours are extracted on the plane $z/d = 0$.

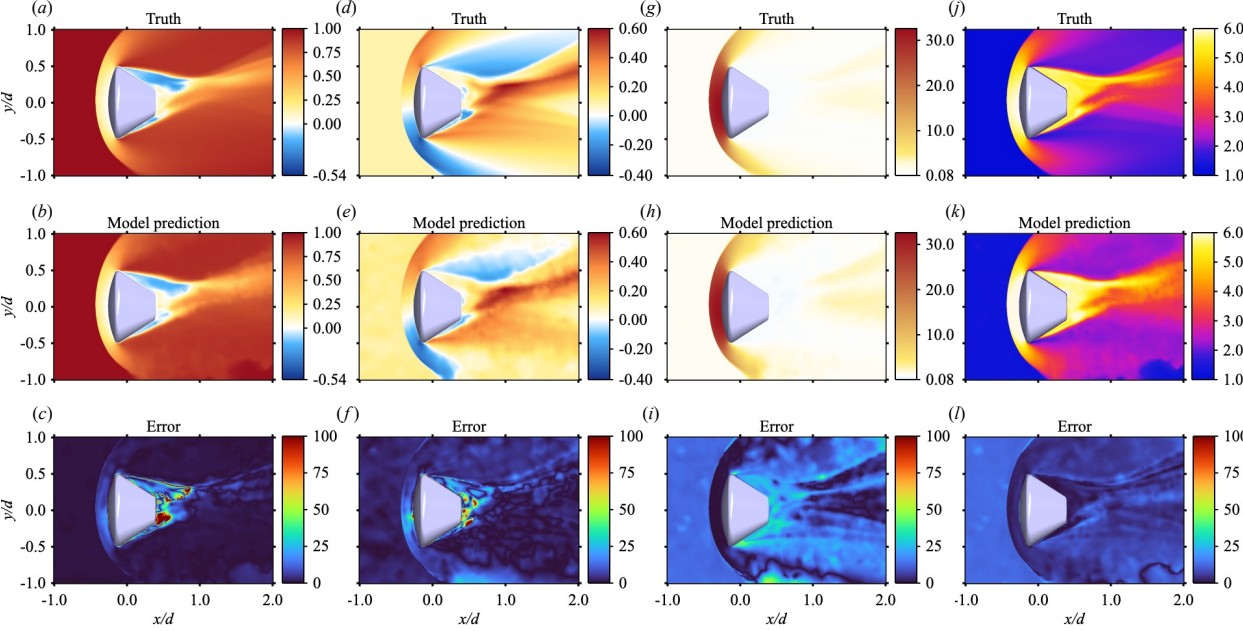

Figure 16: Comparison of ground truth and (best) model predictions at $\alpha = 5°$ for: horizontal velocity $v_x/U_\infty$ (a, b), vertical velocity $v_y/U_\infty$ (d, e), pressure $p/p_\infty$ (g, h), and temperature $T/T_\infty$ (j, k). The normalized percentage error is shown in (c, f, i, l). All contours are extracted on the plane $z/d = 0$.

Overall, the model predictions at these extrapolated angles of attack are within 20% error in a normalized $\mathcal{L}_2$-norm error sense, with the largest discrepancies on the $v_z$ velocity. The most accurate predicted state is pressure with error below 5%, followed by $v_x$ velocity (below 10%) and $v_y$ velocity and temperature $T$ in the range 10-15%. Notably, the model learns the dependency between the orientation of the wake and the

angle of attack also in the extrapolation range of angle of attack. Both at low and high incidence the overall structure of the wake is well predicted as seen from the $v_x$ and $v_y$ velocity fields.

Compared to the interpolation $\alpha$-range predictions (the figures in the main text and in Appendix B.1) the shock layer behind the bow shock presents larger error spots, especially near the stagnation point where the flow comes to rest and acquires the largest values of internal energy and peak temperature roughly 6 times the free-stream.

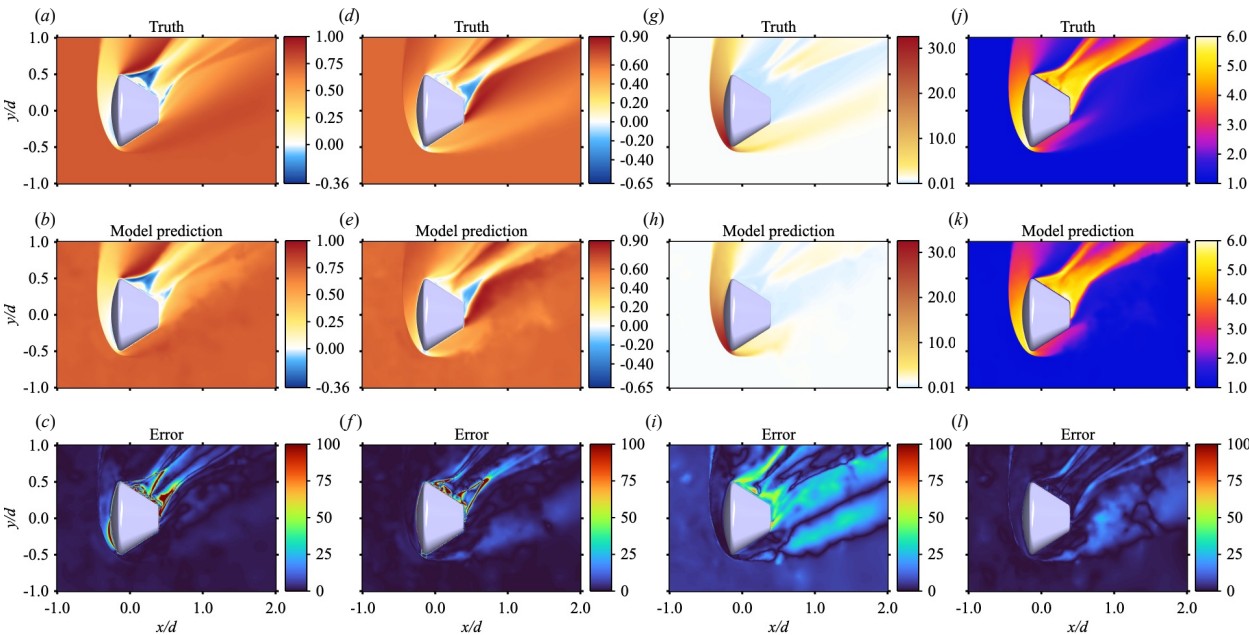

Figure 17: Comparison of ground truth and (best) model predictions at $\alpha = 40°$ for: horizontal velocity $v_x/U_\infty$ (a, b), vertical velocity $v_y/U_\infty$ (d, e), pressure $p/p_\infty$ (g, h), and temperature $T/T_\infty$ (j, k). The normalized percentage error is shown in (c, f, i, l). All contours are extracted on the plane $z/d = 0$.

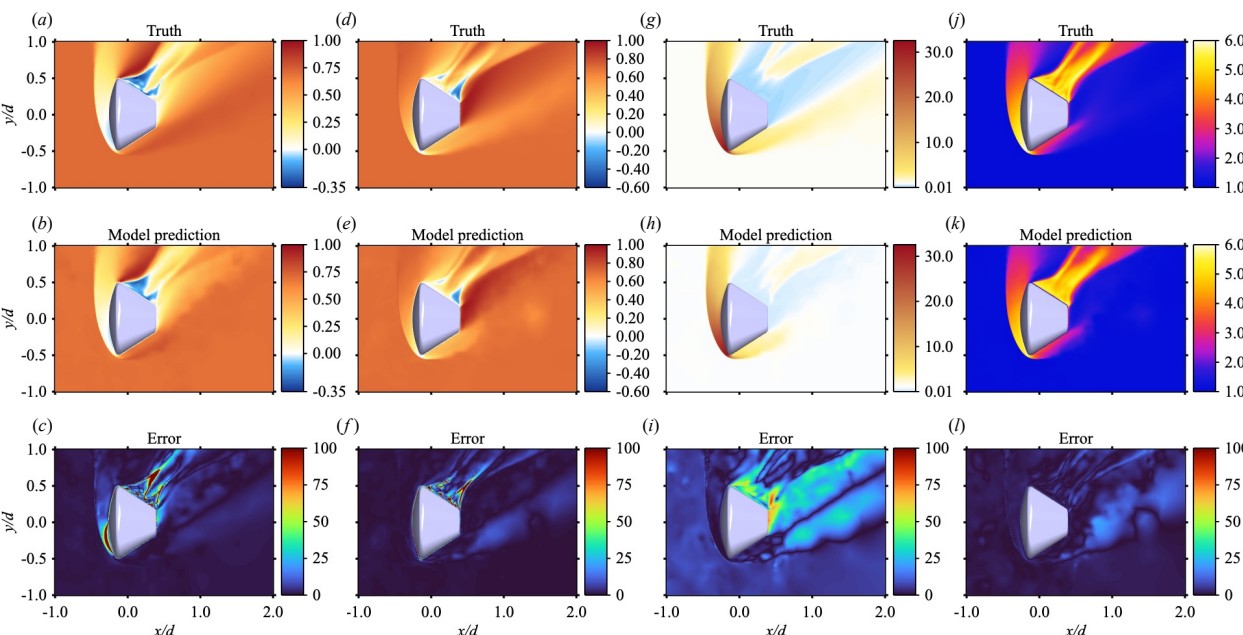

Figure 18: Comparison of ground truth and (best) model predictions at $\alpha = 45°$ for: horizontal velocity $v_x/U_\infty$ (a, b), vertical velocity $v_y/U_\infty$ (d, e), pressure $p/p_\infty$ (g, h), and temperature $T/T_\infty$ (j, k). The normalized percentage error is shown in (c, f, i, l). All contours are extracted on the plane $z/d = 0$.

