# OpenReview forum: "Learning 3D Hypersonic Flow with Physics-Enhanced Neural Fields: A Case Study on the Orion Reentry Capsule"
_TMLR — Decision pending for TMLR_

### Review · Reviewer_SRK4 · 2026-04-24

**Summary Of Contributions:**

This paper proposes a neural field-based method for learning 3D hypersonic flow and tests the method on the Orion reentry capsule environment. The neural field method can speed up hypersonic flow modeling by orders of magnitude compared to traditional computational fluid dynamics methods. The paper further integrates Fourier positional feature embeddings to capture high-frequency changes in the output and a handcrafted heuristic method to refine the neural network output to satisfy the no-slip boundary condition and isothermal wall boundary condition. Empirical results show superior performance compared to alternative methods such as GNN.

**Strengths**
1. The investigated problem of 3D hypersonic flow modeling is important and has great application value.
2. The integration of Fourier positional feature embeddings is well-motivated and shows clear empirical benefits.

**Weaknesses**
1. There is no fundamental methodological novelty in this paper. It is a direct application of the neural field method to hypersonic flow modeling.
2. Though claimed as an "aerothermodynamic simulator," the proposed method does not take time or the current temperature, pressure, and velocity fields as input, which means the network only models a fixed field rather than a time-dependent simulation. Additionally, the network cannot consider any varied initial conditions other than the angle of attack $\alpha$.
3. The empirical results are not sufficient. Only the GNN method is compared as a baseline, and experiments are conducted on only a single environment (Orion reentry capsule). Given that this environment also involves many simplifications such as mesh simplification and the laminar flow assumption, the experiments are not sufficient to demonstrate the effectiveness of the proposed method.

**Audience:**

Yes

**Audience Explanation:**

The paper focuses on hypersonic flow modeling, which has significant real-world applications.

**Broader Impact Concerns:**

None.

**Claims And Evidence:**

No

**Claims Explanation:**

Despite Weakness 2 mentioned above, the paper also makes some misleading claims about previous methods. For example, the paper claims that the neural field approach is superior compared to neural operator as neural operator require a large number of training instances. However, neural field-based methods would also require a large number of instances if generalization to new settings is expected. This paper only considers generalization over the one-dimensional angle $\alpha$ and not over initial fields or boundary conditions, which makes the problem easier than the settings commonly considered in neural operator papers.

**Requested Changes:**

1. (Critical) In the experiments, to make it a real dynamics simulator, the network should take in the current pressure, temperature, and velocity fields as input, or include time $t$ as input to make it a time-dependent dynamics simulator.
2. (Critical) To strengthen the empirical results, comparisons with more baselines should be included, and experiments should be conducted on more environment settings such as different meshes.

Also see the weaknesses mentioned above.

---

> ### Author Response · Authors · 2026-06-01
>
> We thank Reviewer SRK4 for the careful review and for recognizing the importance of 3D hypersonic flow modeling and the empirical value of Fourier positional feature embeddings. We have revised the manuscript to address the reviewer’s concerns, especially regarding the scope of the proposed model, the terminology used to describe it, and the breadth of the empirical evaluation.
>
> **Weakness 1: Methodological novelty**
>
> We understand the reviewer’s concern that the paper builds on coordinate-based neural fields rather than introducing a fundamentally new neural architecture. Our intended contribution is not a new general-purpose neural-field architecture, but rather the formulation, implementation, and evaluation of physics-enhanced neural fields for a large-scale 3D hypersonic aerothermodynamic problem. In particular, the paper contributes: (i) a large-scale 3D CFD case study for the Orion reentry capsule with independently adapted meshes across AoA; (ii) a coordinate-based surrogate formulation that can learn directly from these irregular, AoA-dependent CFD point clouds; (iii) Fourier feature and boundary-condition ablations for hypersonic shock-dominated flows; and (iv) an aerothermodynamic evaluation beyond standard ML losses, including shock-gradient profiles and surface pressure coefficient distributions. In particular point (iv) and discussions and insights such as those presented in Section 8.1 (Challenges capturing sharp discontinuities in hypersonic flows) are often lacking in the ML literature and can be of great interest to aerodynamicists transitioning into AI4Science.
>
> **Weakness 2: “Aerothermodynamic simulator” versus steady surrogate**
>
> We agree that the original wording could be misleading. The proposed model does not perform time-dependent rollout and does not take the current pressure, temperature, or velocity field as input. It is therefore not a dynamics simulator in the sense of predicting temporal evolution from an initial condition. Instead, it is a steady, parametric aerothermodynamic surrogate: it maps spatial coordinates and angle of attack to the corresponding steady CFD solution fields.
>
> We have revised the manuscript accordingly. In particular, we now use “surrogate” and “steady hypersonic flow predictor” language throughout the paper, and we explicitly state that the dataset consists of steady-state flow fields defined over spatial coordinates and AoA, not time-dependent trajectories. We also clarify that the model generalizes over the operating parameter considered in this study, namely AoA for a fixed capsule geometry, and does not claim generalization over arbitrary initial conditions, boundary conditions, or temporal dynamics.
>
> We have added Section 3 Preliminaries for Compressible Laminar Flow Simulations in which we discuss why aerodynamicists and engineers use this particular type of simulations and the motivation behind them.

---

> > ### Author Response · Authors · 2026-06-01
> >
> > **Weakness 3: Empirical scope, baselines, and single environment**
> >
> > We agree that the original empirical evaluation was too limited, and we have expanded it in the revised manuscript.
> >
> > First, we added additional baselines. Beyond the original GNN comparison, we now include additional GNN variants and classical ML baselines, including linear regression, ridge regression, k-NN regression, and random forests (Table 2 and 3). These results are reported in the revised experimental section and help contextualize the neural-field performance.
> >
> > Second, we added additional generalization experiments. The original validation split only evaluated interpolation across AoA. We now include High-AoA extrapolation, Low-AoA extrapolation, and Sparse-AoA interpolation experiments. These additional splits provide a clearer picture of the model’s ability to generalize across the AoA parameter. See Table 5 and visualizations in Appendix B.2 Flow prediction for extrapolation results.
> >
> > Third, we added coarse-to-fine mesh experiments. In these experiments, the model is trained on a coarser subset of spatial points and evaluated on held-out finer spatial points. This does not constitute a new physical geometry, but it does test whether the coordinate-based representation transfers across different spatial discretizations. See Table 6.

---

> > > ### Author Response · Authors · 2026-06-01
> > >
> > > **Requested Changes:**
> > >
> > > *(Critical) In the experiments, to make it a real dynamics simulator, the network should take in the current pressure, temperature, and velocity fields as input, or include time as input to make it a time-dependent dynamics simulator.*
> > >
> > > We have changed the terminology from simulator to surrogate and explain the type of simulations we are conducting and the motivation behind them in 3 Preliminaries for Compressible Laminar Flow Simulations. We believe this provides enough aerodynamics background for the reader to understand the scientific problem at hand. In practice, in engineering and in industry is common to use steady-state flow field predictions rather than time-varying ones which are more common for simpler geometries and in academic toy examples.
> > >
> > > *(Critical) To strengthen the empirical results, comparisons with more baselines should be included, and experiments should be conducted on more environment settings such as different meshes.*
> > >
> > > We have added the following baselines GIN, ChebNet, SGC, linear regression, ridge, KN regressor, random forest, GATv2 see Table 2 and 3.
> > >
> > > We have added experimental setups with extrapolation and interpolation across angle of attack and coarse to fine mesh experiments in Section 7.4 Additional generalization experiments: interpolation and extrapolation across angle of attack and
> > > coarse to fine meshes.
> > >
> > > We have also added plots of how the meshes and the graphs change as a function of angle of attack in Figure 4 and Figure 5, to make the variation in mesh geometry more explicit.

---

### Review · Reviewer_w7Y7 · 2026-05-02

**Summary Of Contributions:**

This paper proposed a coordinate-based neural field method for computational fluid dynamic surrogate. Under an orion reentry capsule dataset with different attack of angels, the method shows its ability in accurately predicted fluid field around the reentry capsule much more faster than numerical method. The model is enhanced with Fourier positional feature mappings to capture sharp discontinuities in hypersonic flows, and with soft boundary condition enforcement to encourage physically plausible behavior near the wall. The authors also compare the proposed neural field against GNN-based surrogates and provide an aerothermodynamic analysis of the predictions at unseen angles of attack.

**Strength:**
1. This work demonstrates the neural field approach on a full-scale Orion capsule with around 130M nodes CFD meshes per AoA. Showing that coordinate-based neural fields remain trainable and effective at large scale data with few samples.

2. This work softly incorporates no-slip and isothermal wall conditions into the neural field through a simple multiplicative gating motivated by fluid dynamics principles, and the ablation in Table 4 confirms that this brings a substantial improvement on unseen AoAs.

3. This work goes beyond standard ML metrics and evaluates the model under a range of aerodynamic metrics, such as shock gradient profiles and surface pressure coefficient distributions, providing a more comprehensive perspective that is convincing to the domain audience.

**Weakness:**

1. This paper trains on only 8 angles of attack and validates on 2 unseen angles (15, 30), both of which fall **within** the range of training angles. The evaluation is therefore interpolation-only, yet a broader claim of extrapolation require more evidence to support.

2. This paper only considers a single Orion capsule geometry, while one of the central motivations of CFD-based design is geometric optimization and exploration. Without evaluating the model's ability to generalize across geometries, it is hard to assess whether the proposed neural field is a practical surrogate for the design workflow envisioned in the introduction and conclusion.

3. This paper includes two types of baselines: GNN-based surrogates and a discussion of neural operators. However, the GNN baselines are built on k-NN graphs rather than the original CFD mesh. And the neural operator in Section 5.1 is not accompanied by any comparison with FNO / DeepONet / GINO-type baselines.

**Audience:**

Yes

**Audience Explanation:**

Yes. The findings of this paper would be of interest to several communities including AI4Science and AI4Engineering researchers working on AI surrogates for physical simulation and the CFD and aerodynamics community exploring data-driven alternatives to numerical solvers.

The 3D hypersonic case study on a realistic large-scale geometry (Large in single sample but small in sample size) also provides a useful examples for current AI4S community.

**Broader Impact Concerns:**

No broader impact concerns.

**Claims And Evidence:**

Yes

**Claims Explanation:**

Yes, partially.

The claims in this paper regarding the proposed neural field method itself are well supported. The authors provide thorough ablations on each component and clearly demonstrate the effect and meaning of each design choice.

The comparison with alternative surrogates is less convincing. The GNN baseline is built only on a k-NN graph. The justification that mesh-based GNNs would not perform better relies on visual inspection of Figure 3, which is not reliable at this mesh scale. At minimum, a direct visualization of the k-NN graph should be provided if original mesh is unavailable. The neural operator discussion in Section 5.1 also overlooks recent methods such as GINO that handle 3D geometries beyond grid fields. Also the claimed speedup compares only inference time against CFD wall time, while a fairer comparison should also account for training cost.

**Requested Changes:**

1. Provide a visualization of the k-NN graph used for the GNN baselines.

2. Report training time cost to comprehensively evaluate total speedup.

3. To support the claim of exploration ability, it would be useful to retrain the model on AoAs {0°, 5°, 10°, 20°, 25°, 35°} and evaluate on held-out higher angles for extrapolation, as well as on angles that do not fall on the 5° grid (e.g., 28°, 41°) to test interpolation ability.

4. Given that recent methods such as GINO (Geometry-Informed Neural Operator for Large-Scale 3D PDEs) are designed for unstructured 3D geometries, the claim in Section 5.1 that "neural operator architectures are most convenient on structured discretizations" should be revised. And good to add comparison with GINO.

5. The reference "Semi-supervised classification with graph convolutional networks" appears twice.

---

> ### Author Response · Authors · 2026-06-01
>
> We thank Reviewer w7Y7 for the careful and constructive review. We appreciate the positive assessment of the large-scale 3D hypersonic case study, the physics-enhanced neural-field formulation, and the aerothermodynamic analysis. We have revised the manuscript to address the reviewer’s concerns, as detailed below. We have uploaded the revised manuscript, see pdf.
>
> **Weakness 1: Interpolation-only evaluation over AoA**
>
> We agree with the reviewer that the original validation protocol evaluated only interpolation in angle of attack, since the held-out angles (15^\circ) and (30^\circ) lie within the range of the training angles. We have therefore added additional experiments to explicitly assess both interpolation and extrapolation.
> In addition to the original split, we now report the following settings: High-AoA extrapolation, Low-AoA extrapolation, Sparse-AoA interpolation. These experiments clarify the distinction between interpolation and extrapolation and provide a more complete picture of the model’s generalization with respect to AoA. Please see Section 7.4 Additional generalization experiments: interpolation and extrapolation across angle of attack and coarse to fine meshes and Appendix B.2 Flow prediction for extrapolation results. We do not include off grid angles of attack because this would require rerunning very expensive CFD simulations, but there is nothing 'special' about on grid vs off-grid angles, it is just a dataset design choice.
>
> **Weakness 2: Single-geometry limitation**
>
> We agree that generalization across capsule geometries would be highly relevant for aerodynamic design and geometry optimization. The present work focuses on a fixed Orion capsule geometry and should therefore be interpreted as a first step toward neural surrogates for large-scale 3D hypersonic aerothermodynamics, rather than as a complete design-optimization surrogate.
>
> We have revised the manuscript to make this limitation explicit. In particular, we now clarify that the model demonstrates generalization across AoA and across independently adapted CFD discretizations, but not across different physical vehicle geometries. The adaptive meshes change substantially with AoA because shock location, wake topology, and refinement regions vary with incidence; Figure 4 (in Section 4.5 Angle-of-attack-dependent mesh adaptation) has been added to make this mesh variation more explicit. This demonstrates that the coordinate-based neural field does not require a fixed grid or a fixed node set across simulations, but it does not replace true geometry generalization.
>
> To further probe discretization generalization, we have also added experiments in which the model is trained on coarser mesh samples and evaluated on finer mesh samples (7.4 Additional generalization experiments: interpolation and extrapolation across angle of attack and coarse to fine meshes). This is closer to a super-resolution or mesh-transfer setting, and it helps assess whether the learned field representation remains consistent across different spatial discretizations. We now present these results as evidence of mesh/discretization flexibility, while clearly identifying cross-geometry generalization as future work.

---

> ### Author Response · Authors · 2026-06-01
>
> **Weakness 3: Baselines and neural operators**
>
> We thank the reviewer for pointing this out. We have revised the baseline discussion in several ways.
> First, we agree that the GNN baselines should not be interpreted as definitive evidence against all mesh-based GNN surrogates. Since the original CFD cell adjacency is not available from the commercial CFD workflow used to generate the data, we cannot perform a controlled comparison against GNNs using the original solver mesh connectivity. We have therefore revised the text to clarify that our GNN experiments are proximity-graph baselines based on k-NN connectivity, not mesh-connectivity GNN baselines. We have also added a direct visualization of the k-NN graph construction, as requested (Figure 5).
>
> Second, we have added additional baselines, including further GNN variants and classical ML baselines, to better contextualize the neural-field results (see Table 2 and Table 3).
>
> Third, we agree that our previous statement about neural operators was too broad. In the original manuscript, we wrote that neural operator architectures are most convenient on structured discretizations. This is accurate for classical FFT-based FNO implementations, but not for recent geometry-aware neural operators such as GINO, which are explicitly designed for irregular 3D geometries and unstructured point-cloud representations. GINO combines graph neural operator components with Fourier neural operator blocks on a latent grid, and was proposed specifically for large-scale 3D PDE and CFD settings .
>
> We have revised Section 6.1 to distinguish classical FNO-style architectures from geometry-informed neural operators such as GINO. We now discuss GINO as an important related method and clarify why our experimental focus differs. In particular, the present paper studies a fixed physical geometry with a small number of expensive CFD realizations and AoA as the main conditioning parameter. By contrast, GINO is primarily motivated by operator learning across large ensembles of varying geometries and/or boundary conditions. For example, the GINO paper evaluates on datasets with hundreds of vehicle geometries, whereas our current dataset contains one Orion capsule geometry with multiple AoA-dependent CFD solutions .
> For this reason, we believe a full GINO comparison is an important direction for future work, especially for geometry-varying capsule datasets, but is not directly aligned with the central experimental regime of the current paper. We have softened the claims accordingly and now state that geometry-aware neural operators are relevant alternatives rather than being inapplicable.
> We also clarify that the difference is partly due to the CFD workflow required by the physical regime. The GINO benchmarks are based on comparatively standard external-aerodynamics vehicle simulations, such as Ahmed-body and car geometries, using OpenFOAM-based workflows. OpenFOAM is well suited to many such conventional CFD settings and, importantly for learning on meshes, makes it possible to access and export mesh information. Our setting is substantially different: we simulate the Orion reentry capsule at hypersonic speed, where accurate resolution of bow shocks, shock/boundary-layer structure, and strong localized gradients requires advanced shock-adapted mesh refinement and robust hypersonic CFD setup. In our work this was achieved using STAR-CCM+, whose adapted final mesh connectivity is not readily exportable in the form needed for mesh-based GNN or GINO-style baselines. Therefore, the absence of original-mesh GNN/GINO baselines is not simply an implementation choice, but a consequence of the CFD data-generation pipeline required for this hypersonic regime.

---

> ### Author Response · Authors · 2026-06-01
>
> **Requested changes**
>
> *1. Provide a visualization of the k-NN graph used for the GNN baselines.*
>
> We have added a visualization of the k-NN graph construction. This makes the GNN baseline more transparent (Figure 5)
>
> *2. Report training time cost to comprehensively evaluate total speedup.*
>
> We have added the training cost (Section 7.5 Neural inference time). The originally reported speedups referred only to post-training inference time compared with CFD wall time.
>
> *3. Add extrapolation and additional interpolation experiments.*
>
> We have added the extrapolation and sparse-interpolation experiments described above, including held-out high-AoA and low-AoA regimes, as well as interpolation from a reduced set of training AoAs.
>
> *4. Revise the neural-operator discussion and address GINO.*
>
> We have revised Section 6.1 to correct the overbroad statement about neural operators and structured discretizations. The revised text now explicitly discusses GINO and related geometry-aware neural operators. We clarify that GINO is relevant to the broader problem of 3D CFD surrogate modeling, while also explaining that our present setting is a fixed-geometry, low-number-of-CFD-realizations regime where coordinate-based neural fields are particularly natural.
>
> *5. Remove duplicate reference.*
> We have removed the duplicate reference to “Semi-supervised classification with graph convolutional networks.”
>
> Thank you again for the constructive suggestions. We believe the added experiments, revised baseline discussion, k-NN visualization, and training-cost analysis substantially improve the manuscript.

---

### Review · Reviewer_KbQz · 2026-06-08

**Summary Of Contributions:**

Contribution: This paper develops a neural surrogate for high-speed 3D flow around the Orion reentry capsule using Fourier positional encoding features and enforced boundary conditions. Experiments against classical baselines show improved predictions near critical regions such as shocks and boundaries, together with substantial speedups over their computational fluid dynamics (CFD) counterparts.


Strengths:

1. The paper addresses a difficult and practically relevant problem by learning 3D hypersonic flow fields on unstructured meshes.
2. The 3D dataset for hypersonic flow field and the detailed explanation of the setup, mesh refinement and physical assumptions is a plausible contribution.
3. The evaluation standards are physically meaningful as they go beyond overall prediction and error and examine critical quantities such as shock profile, boundary layers and pressure distribution.
4. The presentation is clear and readable.


Weaknesses:

1. A primary concern is that the work does not appear to introduce a sufficiently novel machine learning method. The proposed model mainly combines already established components: a coordinate-based MLP with Fourier features, with analytic equations boundary conditions for which the theoretical explanation is not very clear. The work is solid, but its main novelty lies in the hypersonic CFD application and dataset rather than in the machine learning methodology, which limits its contribution to a general ML journal.

2.  The baseline comparison is not strong enough to support the broad conclusion that GNNs are unsuitable for hypersonic flow prediction. The experiments use k-NN graphs and mostly standard message-passing architectures rather than the original mesh connectivity or more specialized multiscale models.

3. The term “physics-enhanced” may be too broad, since the model only enforces selected wall boundary conditions and does not include conservation laws or PDE residuals.

**Additional Comments:**

The paper is technically solid and the application is interesting. However, the objective and intended contribution are not framed clearly enough. It is difficult to determine whether the paper aims to introduce a new ML methodology, establish the suitability of neural fields for hypersonic flow, or provide an application-focused empirical study. The authors should clarify the main research question and ensure that the claims are aligned with that objective.

**Audience:**

No

**Audience Explanation:**

Yes. Researchers working on scientific machine learning, neural fields, and data-driven CFD surrogates may be interested in the empirical findings, particularly the application to large 3D hypersonic flow fields, the effect of Fourier features near shocks, and the comparison with GNN baselines. However, the relevance is mainly application-driven, and the paper offers limited novelty in terms of general machine-learning methodology.

**Claims And Evidence:**

Yes

**Claims Explanation:**

Yes. The main empirical claims are supported by clear experiments, including ablations on Fourier features and boundary-condition enforcement, comparisons with several baselines, and physically meaningful evaluations of shocks, boundary layers, and surface pressure. The reported results are generally consistent and convincing for the specific Orion geometry and operating conditions considered.

However, some broader claims should be stated more carefully. In particular, the conclusions about the general unsuitability of GNNs and the general applicability of the framework to 3D hypersonic aerothermodynamics are stronger than the current evidence, since the experiments use one fixed geometry, a limited range of flow conditions, and mostly standard k-NN-based GNN baselines.

**Requested Changes:**

1. The paper should more clearly separate the novelty of the CFD dataset and application from the novelty of the machine-learning method. This is critical, since the model mainly combines established components such as a coordinate-based MLP, Fourier features, and analytic boundary-condition enforcement.

2. The baseline comparison should be strengthened. The GNN experiments use k-NN graphs and mostly standard message-passing models.   Stronger baselines could include MeshGraphNets [1] and the Graph U-Net architecture [2][3]. The conclusion should therefore be limited to the specific GNN baselines tested, and should be generalized to GNN's only after the suggested baselines have been applied to the problem.

3. The generalization claims should be narrowed. The experiments use one fixed geometry and one main flow regime, with angle of attack as the primary varying parameter. Testing an additional geometry or operating condition would be needed to support broader claims.

4. The term “physics-enhanced” should be used more carefully. Since the model only enforces selected wall boundary conditions and does not include conservation laws or PDE residuals (such as mass, momentum, energy), “boundary-condition-constrained” may be more accurate.



References

[1] T. Pfaff, M. Fortunato, A. Sanchez-Gonzalez, and P. W. Battaglia, “Learning Mesh-Based Simulation with Graph Networks,” ICLR, 2021.

[2] H. Gao and S. Ji, “Graph U-Nets,” in Proceedings of the 36th International Conference on Machine Learning, pp. 2083–2092, 2019.

[3] S. Yang, R. Vinuesa, and N. Kang, “Enhancing Graph U-Nets for Mesh-Agnostic Spatio-Temporal Flow Prediction,” arXiv preprint arXiv:2406.03789, 2024.

---

> ### Author Response · Authors · 2026-06-16
>
> We thank the reviewer for the careful and constructive assessment of our manuscript.
>
> **Requested Changes**
>
> 1. We have revised the introduction and conclusion to more clearly separate the dataset/application contribution from the machine-learning methodology. In particular, we now explicitly frame the paper as an application-focused empirical study and dataset contribution. For example, we added the following statement: “In summary, this work is best understood as an application-focused empirical study and dataset contribution for 3D hypersonic aerothermodynamic surrogate modeling, presented through a case study on the Orion reentry capsule.”
>
> 2. We have strengthened the baseline comparison by adding the suggested Graph U-Net and MeshGraphNet baselines. In addition, we included GraphGPS, which incorporates global attention. As described in the revised manuscript, “GraphGPS differs from purely message-passing architectures because its global attention mechanism allows each node to aggregate information from all other nodes in the subsampled graph within a layer.” These additional baselines improve over some of the simpler GNN architectures, but still underperform compared with the coordinate-based neural field in our setting. Please see Section 7.2 and Tables 2–3.
>
> 3. We have adjusted the claims in the introduction and conclusion. In addition, the original title itself mentions: 'A Case Study on the Orion Reentry Capsule', which makes the focus explicit.
>
> 4. We have clarified the meaning of “physics-enhanced” in this work. In the contribution list, we now explicitly state that the model focuses on boundary-condition enforcement and does not include conservation laws or PDE residuals: “We propose the use of physics-enhanced 3D neural fields to predict steady hypersonic flow around the geometry (note that we focus on boundary conditions, but do not include conservation laws or PDE residuals).” Also at the end of Section 5, we have included 'Note that the no-slip and isothermal-wall constraints bias the solution toward physically plausible near-wall behavior, but do not constitute a full PDE-constrained formulation.'
>
> Additional Comments:
>
> Regarding the concern that “the objective and intended contribution are not framed clearly enough": the title, introduction, contribution list, and conclusion consistently frame the paper as a case study on the Orion reentry capsule, with the primary contribution being an application-focused empirical study and dataset for 3D hypersonic aerothermodynamic surrogate modeling.
>
> We hope these revisions make the scope and intended contribution of the paper clearer.

---

### Decision · Action_Editor_t3W4 · 2026-07-09

**Recommendation:** Accept as is

**Audience:**

Yes

**Audience Explanation:**

The work presents ML applications in challenging fluid flow modeling tasks and this is an emerging and an interesting area with a lot of potential.

**Claims And Evidence:**

Yes

**Claims Explanation:**

The empirical evaluations are quite strong and are indeed the strength of this paper.